# Deletion of the Circadian Clock Gene *Per2* in the Whole Body, but Not in Neurons or Astroglia, Affects Sleep in Response to Sleep Deprivation

Katrin S. Wendrich [1], Hamid Azimi [2], Jürgen A. Ripperger [1], Yann Ravussin [3], Gregor Rainer [2] and Urs Albrecht [1,*]

1 Department of Biology, University of Fribourg, 1700 Fribourg, Switzerland
2 Section of Medicine, Department of Neuroscience, University of Fribourg, 1700 Fribourg, Switzerland
3 Section of Medicine, Department of Endocrinology, Metabolism and Cardiovascular System, University of Fribourg, 1700 Fribourg, Switzerland
* Correspondence: urs.albrecht@unifr.ch

**Abstract:** The sleep–wake cycle is a highly regulated behavior in which a circadian clock times sleep and waking, whereas a homeostatic process controls sleep need. Both the clock and the sleep homeostat interact, but to what extent they influence each other is not understood. There is evidence that clock genes, in particular *Period2* (*Per2*), might be implicated in the sleep homeostatic process. Sleep regulation depends also on the proper functioning of neurons and astroglial cells, two cell-types in the brain that are metabolically dependent on each other. In order to investigate clock-driven contributions to sleep regulation we non-invasively measured sleep of mice that lack the *Per2* gene either in astroglia, neurons, or all body cells. We observed that mice lacking *Per2* in all body cells (*Per2*<sup>Brdm</sup> and T*Per2* animals) display earlier onset of sleep after sleep deprivation (SD), whereas neuronal and astroglial *Per2* knock-out animals (N*Per2* and G*Per2*, respectively) were normal in that respect. It appears that systemic (whole body) *Per2* expression is important for physiological sleep architecture expressed by number and length of sleep bouts, whereas neuronal and astroglial *Per2* weakly impacts night-time sleep amount. Our results suggest that *Per2* contributes to the timing of the regulatory homeostatic sleep response by delaying sleep onset after SD and attenuating the early night rebound response.

**Keywords:** astrocytes; clock; sleep deprivation; metabolism; sleep regulation

## 1. Introduction

Sleep is a periodically occurring behavior of rest and lack of interaction with the environment. It is a highly regulated state, involving a homeostatic process regulating the increase for readiness to fall asleep during wakefulness and the decrease of sleep intensity during sleep, and a circadian process that schedules sleep and wakefulness to the appropriate time within one day [1].

The circadian system is an endogenous oscillator orchestrating the daily rhythms of biochemistry, physiology, and behavior of most organisms and aligning an organism's inner clock to the external environment. Internal synchronization of the body clocks is achieved by the suprachiasmatic nuclei (SCN) of the hypothalamus [2]. However, instead of a 'master' sleep homeostat, comparable to the 'master' clock in the SCN, sleep homeostasis seems to depend on local and use-dependent processes. The dynamics of sleep homeostasis appear to be genetically determined [3], implying that the sleep homeostat has a molecular substrate. A disruption of cellular clocks and/or destruction of the synchronization between the clocks, as experienced, for instance, in jet-lag and shift-work conditions, affects normal brain function and can lead to sleep disturbances, metabolic problems, and accelerated neurological decline (reviewed in [4]). Mutations in circadian clock genes are associated

with sleep disturbances [5–13] and obesity [14]. Conversely, sleep deprivation can alter the expression [15,16] and DNA binding activity of core clock genes [17], demonstrating a bidirectional relationship between the circadian clock and sleep. Intriguingly, a similar bidirectional relationship also exists between metabolism and sleep. Sleep duration is associated with metabolic syndrome [18] and mistimed food intake most likely affects sleep. Therefore, sleep and the circadian clock may be linked, involving, at least in part, metabolism [19].

Since the *Per2* gene is part of both the circadian clock mechanism [20] as well as the regulatory mechanism of metabolic processes [21,22], mice with a mutation in the *Per2* gene were investigated for its role in sleep regulation [7,23]. These studies indicated that in these mutants not only the circadian component of sleep was altered but also the homeostatic component was affected. In those studies, *Per2* was deleted in the whole organism not differentiating the role of *Per2* in specific cell types. Because sleep is generally viewed as a behavior emerging in the brain and *Per2* is expressed in both neurons and astroglia, which metabolically depend on each other [24], we wanted to test whether neurons (nestin-positive cells) or astroglial cells (gfap-positive cells), were responsible for the described sleep regulatory function of *Per2*. Using a non-invasive method, we analyzed the sleep–wake parameters of total *Per2* (T*Per2*), neuronal *Per2* (N*Per2*) [25], and astroglial *Per2* (G*Per2*) [26] knock-out (KO) animals and compared them with mice containing a *Per2* gene with an in-frame deletion (*Per2$^{Brdm}$*) [27].

## 2. Results

To examine the role of the clock gene *Per2* in sleep including, but not limited to, brain cell specificity, we investigated mice with a mutation (*Per2$^{Brdm}$*) or deletion of this gene in all cells of the body (T*Per2*) or specifically in neurons (N*Per2*) or astroglial cells (G*Per2*), respectively. Due to different *Per2* deletion strategies, *Per2$^{Brdm}$* and T*Per2* mice were not on the same genetic background. To control for this, control animals used in each comparison, termed wild-type (*wt*), were littermates (for N*Per2* and G*Per2*) or corresponding strain wildtypes (mixed SV129/B6 background for *Per2$^{Brdm}$*, B6 background for T*Per2*). Therefore, the *wt* in each of the following comparisons are not the same but specific controls to the corresponding knock-out (*Per2$^{Brdm}$*, T*Per2*, N*Per2* and G*Per2*). Both total knock-out strains were analyzed, because *Per2$^{Brdm}$* mutants have been widely used to study the role of *Per2* in circadian rhythms and T*Per2* mice possess the same genetic background as N*Per2* and G*Per2* animals.

### 2.1. Per2$^{Brdm}$ Mice Display an Altered Sleep–Wake Pattern, Earlier Sleep Onset after SD, and a Temporally Differential Recovery Response after SD

Mice with an in-frame deletion in the *Per2* gene were analyzed for sleep in the PiezoSleep system. These mice express a truncated unstable PER2 protein (*Per2$^{Brdm}$* [28]) that does not translocate into the nucleus [29], and hence behave like a *Per2* knock-out on the transcriptional level. We collected 7 days of baseline sleep data before we applied 6 h of sleep deprivation (SD) starting at light onset. Subsequently, we monitored recovery sleep (Figure 1A). The 7 days of baseline sleep were averaged and the distribution and amount of sleep are shown in Figure 1B (0–24 h). Significant differences between *Per2$^{Brdm}$* mutants and their controls were seen just before the light to dark transition, with the main active phase of *Per2$^{Brdm}$* mice starting earlier than controls, which has also been observed in a previous study [7]. In the first few hours of the dark phase and 2–3 h before the dark to light transition, *Per2$^{Brdm}$* mice were significantly less asleep than controls.

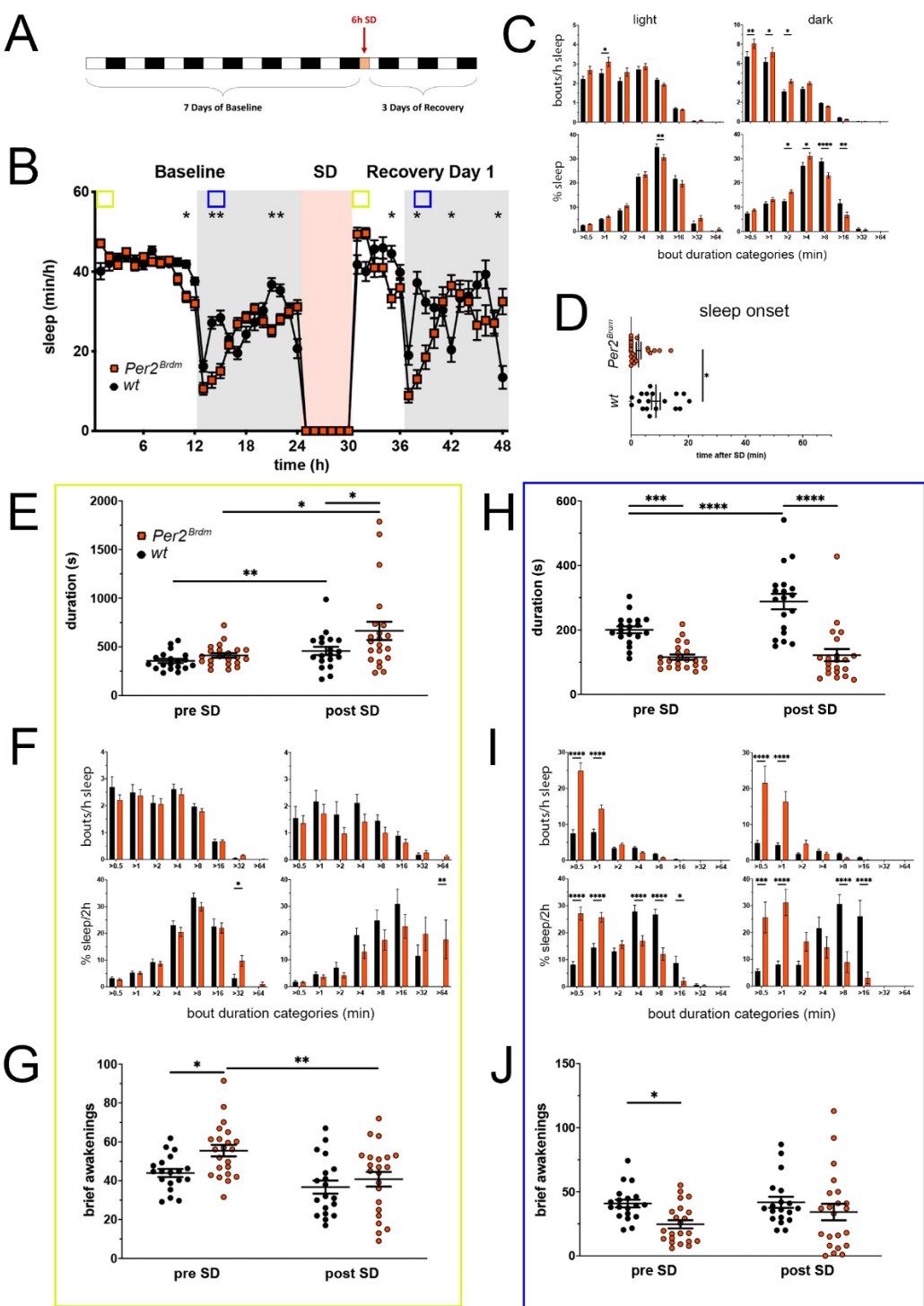

**Figure 1.** Sleep–wake analysis of *Per2^{Brdm}* mutant mice (red) compared to wild type controls (black). (**A**) Scheme of sleep–wake recordings during undisturbed conditions (baseline = BL), sleep deprivation (SD), and sleep–wake recordings during recovery from SD; (**B**) time course of mean (±SEM) hourly values of sleep over 7 days summarized over 24 h baseline. White areas depict 12 h of light and gray shaded areas 12 h of darkness. The rose shaded area depicts 6 h of sleep deprivation (SD) in the light portion. The green and blue squares depict pre- and post-SD time windows being analyzed in detail in E-G (green square, 2 h light phase) and H-J (blue square, 2 h dark phase). Statistically significant differences of sleep between the two genotypes by two-way ANOVA with Šídák's post-test is depicted with an asterisk (* $p < 0.05$, ** $p < 0.01$, $n = 19$ *wt*, $n = 22$ *Per2^{Brdm}*); (**C**) distribution of sleep

bout durations in the light (left panels) and dark phase (right panels) of baseline sleep (hours 0–24 in (**B**)). The top panels show the number of bouts per hour of sleep in each sleep bout category in the light and dark phase. The bottom panels show the percentage of sleep in each of the categories. Values are shown as mean ($\pm$SEM). Two-way ANOVA with Šídák's post-test revealed statistically significant differences between the two genotypes with * $p < 0.05$, ** $p < 0.01$, **** $p < 0.0001$, $n = 19$ *wt*, $n = 22$ *Per2$^{Brdm}$*; (**D**) sleep onset after SD. The *Per2$^{Brdm}$* mutant mice show a significantly earlier onset compared to wild type controls. Mann–Whitney test with * $p < 0.05$, $n = 18$ *wt*, $n = 22$ *Per2$^{Brdm}$*; (**E**) average sleep bout duration pre-SD (2 h in BL from ZT0 to ZT2, hours 1–2 in (**B**)) and post-SD (2 h during recovery after sleep onset). Sleep bout duration is significantly increased in *Per2$^{Brdm}$* mutant mice post-SD. Two-way ANOVA with Šídák's post-test with * $p < 0.05$, $n = 19$ *wt*, $n = 21$ *Per2$^{Brdm}$*; (**F**) sleep bout durations pre-SD (left panels, 2 h in BL from ZT0 to ZT2, hours 1–2 in (**B**)) and post-SD (right panels, 2 h during recovery after sleep onset). *Per2$^{Brdm}$* mice sleep more than controls pre- as well as post-SD in the >32 min and >64 min sleep bout categories. Two-way ANOVA with Šídák's post-test with * $p < 0.05$, ** $p < 0.01$, $n = 19$ *wt*, $n = 22$ *Per2$^{Brdm}$*; (**G**) brief awakenings are increased in *Per2$^{Brdm}$* mice pre-SD but not post-SD. Two-way ANOVA with Šídák's post-test with * $p < 0.05$; ** $p < 0.01$. (**H**) average sleep bout duration pre-SD (2 h in BL from ZT13 to ZT15, hours 13–14 in (**B**)) and post-SD (2 h during recovery from ZT13 to ZT15, hours 38–39 in (**B**)). Sleep bout duration is significantly decreased in *Per2$^{Brdm}$* mutant mice pre-SD and post-SD. Two-way ANOVA with Šídák's post-test with *** $p < 0.001$, **** $p < 0.0001$, $n = 19$ *wt*, $n = 21$ *Per2$^{Brdm}$*; (**I**) sleep bout durations pre-SD (left panels, 2 h in BL from ZT13 to ZT15, hours 13–14 in (**B**)) and post-SD (right panels, 2 h during recovery from ZT13 to ZT15, hours 38–39 in (**B**)). *Per2$^{Brdm}$* mice sleep had more short sleep bouts in pre-SD as well as post-SD, * $p < 0.05$, *** $p < 0.001$, **** $p < 0.0001$, but wild-type controls showed a higher sleep percentage in mid and long sleep bouts in pre-SD as well as post-SD. Two-way ANOVA with Šídák's post-test with *** $p < 0.001$, $n = 19$ *wt*, $n = 22$ *Per2$^{Brdm}$*; (**J**) brief awakenings are decreased in *Per2$^{Brdm}$* mice pre-SD, but not post-SD. Two-way ANOVA with Šídák's post-test with * $p < 0.05$, $n = 19$ *wt*, $n = 22$ *Per2$^{Brdm}$*.

The sleep–wake pattern after SD, which took place in the first half of the light phase, differed between the two genotypes. While *Per2$^{Brdm}$* mice slept more right after SD, they became more awake just before the light to dark transition, similar to baseline. The sleep profiles of both genotypes in the dark phase of recovery followed the same dynamics as during baseline, but with more enhanced sleep and wake peaks and as in baseline with the *Per2$^{Brdm}$* mice lagging 4 h behind the control animals in their time-course. Overall, during the light or the dark phase, *Per2$^{Brdm}$* mice had the same amount of sleep as controls and wheel running activity was also similar (Figure S1). In baseline, sleep bout distribution analysis in the light and dark phase revealed that sleep of *Per2$^{Brdm}$* mice contained a higher number of short sleep bouts than controls, especially during the dark phase, indicating that sleep was more fragmented in the *Per2$^{Brdm}$* mice (Figure 1C, upper panels). Accordingly, represented as the percentage of total time spent asleep, *Per2$^{Brdm}$* mice spent less of their sleeping time in longer sleep bouts (>8 min) during the dark phase in comparison to controls. In the light phase, sleep percentages per bout length category were similarly distributed between the two genotypes (Figure 1C, lower panels). In sum, both types of analyses point towards an increased sleep fragmentation in the dark period for *Per2$^{Brdm}$* mice.

Remarkably, after SD, all mice went to sleep within 20 min after the end of SD, but *Per2$^{Brdm}$* mice fell asleep significantly faster than controls (Figure 1D), suggesting a higher intrinsic sleep pressure of mice lacking *Per2*, indicating a role of *Per2* in sleep homeostasis.

To estimate the immediate effects of SD on *Per2$^{Brdm}$* mice, we compared sleep architecture within the two hours of highest sleep pressure between the corresponding undisturbed and sleep-deprived conditions, which are the first two hours after the active phase in baseline (ZT0-2) and the two hours after sleep onset post-SD (ZT6-8 plus time until sleep onset), respectively (indicated by the bright green squares in Figure 1B). Compared to baseline, SD prolonged sleep bout duration in both genotypes. Additionally, compared to controls, *Per2$^{Brdm}$* mice sleep bouts were longer in the two hours of highest sleep pressure during the light phase after sleep deprivation, but not in baseline (Figure 1E). More importantly, there

is an interaction as the relative increase is larger in *Per2Brdm* mice. This would corroborate the shorter sleep onset and a presumed higher build-up of sleep pressure. As another observation next to longer bouts, within these 2-h windows *Per2Brdm* mice also gained more total sleep from pre-SD to post-SD than their controls (Figure S2A). When sorting sleep bouts by their length into duration categories, the number of sleep bouts within each duration category did not differ between the genotypes for both the pre-SD and post-SD condition (Figure 1F, upper panels). However, when comparing how many sleep bouts in each duration category contributed to the overall time spent asleep during these two hours, we observed that *Per2Brdm* mice spent more of their time asleep in longer sleep bouts, which was even more enhanced post-SD (>32 min pre-SD, >64 min post-SD, Figure 1F, lower panels). Within this time window, mutant mice responded to SD with a decrease in sleep bout numbers of short to medium length (for controls only in the very short bout range and trends otherwise) and an increase of time spent asleep in medium to long range (trending for controls) (Figure S3A).

We also counted the number of brief awakenings (<16 s) interrupting sleep, which are considered a marker of sleep fragmentation, and thus inversely correlated with sleep depth [30]. At the time sleep pressure was highest in baseline (ZT0-2), more brief awakenings were observed in *Per2Brdm* mice than in controls, indicating more fragmented sleep under homeostatically unchallenged conditions. Under challenged conditions (SD), on the other hand, and coherent with our previous findings of prolonged average sleep bout length and an increase of time spent asleep in the long bout range after SD, brief awakenings were considerably reduced in mutant mice and comparable to controls (Figure 1G), implying a stronger immediate response of the sleep homeostat. Overall, *Per2Brdm* mice showed more consolidated sleep within this 2-h window of highest sleep pressure after SD than before.

Next, the effects of lack of *Per2* on mouse sleep under undisturbed conditions and after SD were analyzed in detail in the dark phase. Since mice are usually active in the dark phase, this period represents an opportunity to regain more sleep after SD than is possible in the rest phase. Therefore, the analyzed two-hour time window was set to ZT13 to ZT15 (blue squares in Figure 1B), which is early on in the dark phase, while effects of dark onset triggering or de-masking activity are excluded. The duration of an average sleep bout within this 2-h window of the dark phase was shorter in *Per2Brdm* mice pre-SD (2 h in BL from ZT13–ZT15, hours 13–14 in Figure 1B) as well as post-SD (2 h during recovery from ZT13–ZT15, hours 38–39 in Figure 1B). Whereas control mice responded to SD with an increase of average sleep bout length and of total sleep minutes, *Per2Brdm* mice did not react. Furthermore, average sleep bout length and total sleep time was, under both conditions (pre- and post-SD), reduced in mutant mice (Figure 1H and Figure S4A). In contrast to the 2-h time window during the light phase (Figure 1E,F), the number of short sleep bouts in the 2 h slot within the dark phase were significantly increased in *Per2Brdm* mice pre- as well as post-SD (Figure 1I, upper panels). This was accompanied by an increased sleep percentage spent in the short bout bins and a reduced sleep percentage in the longer sleep bout range (Figure 1I, lower panels). Only controls responded to SD with a decrease of short sleep bout numbers and an increase of percentage spent asleep in longer sleep bouts (Figure S5A). In contrast to the light phase, brief awakenings in the dark phase pre-SD were reduced in *Per2Brdm* mice compared to controls, but post-SD no difference between the two genotypes was observed anymore (Figure 1J). Since no effect of SD on the number of brief awakenings was detectable in both genotypes, it can be assumed that sleep splintering at this time window remains unaffected by SD.

Taken together, *Per2Brdm* mice show an earlier activity onset before dark onset and divergent sleep–wake dynamics during the dark phase under baseline conditions. The SD response appears to be altered temporally with mutant mice consolidating sleep stronger within the first two hours after sleep onset following SD, but exhibiting a more fragmented sleep pattern afterwards in the dark period (ZT13-15). The immediate homeostatic response

to SD with faster sleep onset and more consolidated sleep point to a higher accumulated sleep pressure during SD, something that would be attributable to the lack of *Per2*.

*2.2. TPer2 Mice Display Earlier Sleep Onset after SD and More Fragmented Recovery Sleep in the Dark Phase*

Mice with a complete deletion of *Per2* in all body cells [24], designated as T*Per2* KO mice, were investigated for sleep in the PiezoSleep system (Figure 2). Data were collected as described above under 2.1. A significant difference between T*Per2* KO mice and their controls could be seen in the middle of the dark period of baseline sleep at 18 h with more sleep in the KO mice (Figure 2A,B). The rest of the sleep–wake profile looked similar to control animals. The sleep profile in the light phase after SD was equally similar between the two genotypes. Intriguingly, the sleep profile in the dark phase of recovery was almost identical to baseline sleep profiles in the dark period except for hour 39, in which control mice gain significantly more sleep than in baseline. As mentioned before, this attenuated recovery sleep response shortly after dark onset was also discernable in the controls of the *Per2^{Brdm}* data set. Sleep bout distribution analysis in the 12 h light and 12 h dark phase of baseline revealed that sleep of T*Per2* KO mice contained more medium short sleep bouts than that of controls in the dark phase, but not in the light phase. This indicates that sleep was more fragmented in the T*Per2* KO mice during the active/dark phase, but was comparable to controls in the resting/light phase (Figure 2C, upper panels). The distribution of relative sleep amount spent in the eight duration categories in the light phase was similar between the two genotypes, while in the dark phase in T*Per2* KO animals, the sleep percentage was significantly increased in the medium length bout category (>8 min, Figure 2C, lower panels). After SD, sleep onset was significantly faster in T*Per2* KO mice compared to controls (Figure 2D), suggesting that the knock-out mice had a higher build-up of sleep pressure during SD and recovered sleep faster than controls similar to the *Per2^{Brdm}* mutant mice (Figure 1D). Still, average sleep bout duration pre-SD ZT0-2 as well as in the first two hours during the light phase after sleep onset post-SD (between ZT6 and ZT7) was unchanged between T*Per2* KO mice and controls, with the only difference that controls, as a response to SD, significantly lengthened their average sleep bouts post-SD (Figure 2E). Although T*Per2* KO mice gained more sleep during baseline in this time window, sleep amount was equal between genotypes post-SD since controls significantly increased their total sleep time post-SD (Figure S2B). Furthermore, the number of sleep bouts and the percentage of sleep spent in each duration bin were not different between the genotypes, indicating overall a similar quality of sleep (Figure 2F). Consistent with these observations, T*Per2* KO mice did not differ in the number of brief awakenings compared to controls (Figure 2G). The effects of SD were only visible in the controls with an increase of total sleep and a reduction of short sleep bouts during this time (Figures S2B and S3B).

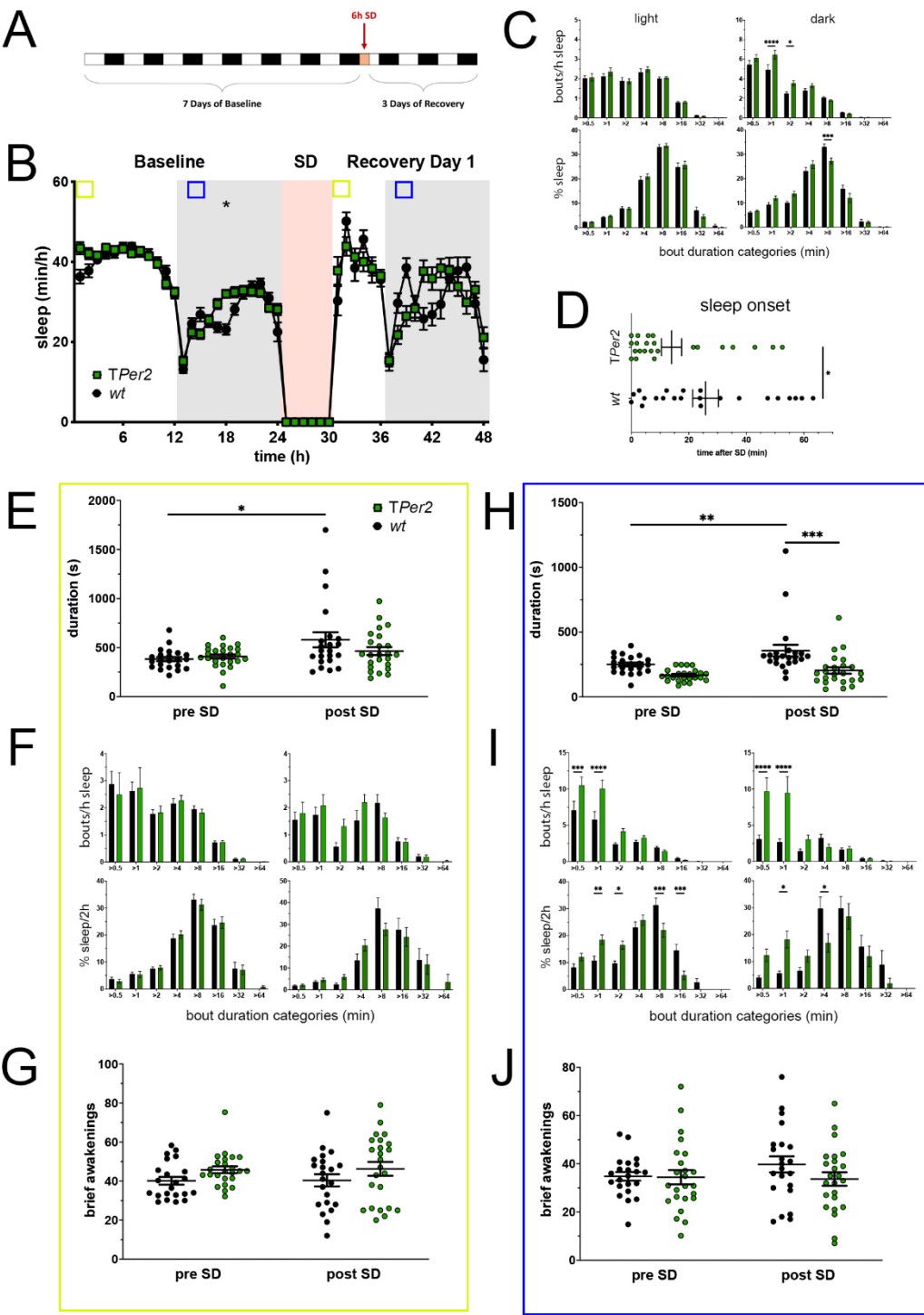

**Figure 2.** Sleep–wake analysis of total *Per2* knock-out (T*Per2* KO) mice (green) compared to wild type controls (black). (**A**) Scheme of sleep–wake recordings during undisturbed conditions (baseline = BL), sleep deprivation (SD), and sleep–wake recordings during recovery from SD; (**B**) time course of mean (±SEM) hourly values of sleep over 7 days summarized over 24 h baseline. White areas depict 12 h of light and gray shaded areas 12 h of darkness. The rose shaded area depicts 6 h of sleep deprivation (SD) in the light portion. The green and blue squares depict pre- and post-SD time windows being analyzed in detail in E-G (green square, 2 h light phase) and H-J (blue square, 2 h dark phase). Statistically significant differences of sleep between the two genotypes by two-way ANOVA with Šídák's post-test is depicted with an asterisk (* $p < 0.05$, $n = 22$ *wt*, $n = 24$ T*Per2*); (**C**) distribution

of sleep bout durations in the light (left panels) and dark phase (right panels) of baseline sleep (hours 0–24 in (**B**)). The top panels show the number of bouts per hour of sleep in each sleep bout category in the light and dark phase. The bottom panels show the percentage of sleep in each of the categories. Values are shown as mean (±SEM). Two-way ANOVA with Šídák's post-test revealed statistically significant differences between the two genotypes with * $p < 0.05$, *** $p < 0.001$, **** $p < 0.0001$, $n = 22$ *wt*, $n = 24$ T*Per2*; (**D**) sleep onset after SD. The T*Per2* KO mice show a significantly earlier onset compared to wild type controls. Mann–Whitney test with * $p < 0.05$, $n = 22$ *wt*, $n = 23$ T*Per2*; (**E**) average sleep bout duration pre-SD (2 h in BL from ZT0 to ZT2, hours 1–2 in (**B**)) and post-SD (2 h during recovery after sleep onset). Sleep bout duration is comparable in both genotypes. Two-way ANOVA with Šídák's post-test, $n = 22$ *wt*, $n = 24$ T*Per2*; (**F**) sleep bout durations pre-SD (left panels, 2 h in BL from ZT0 to ZT2, hours 1–2 in (**B**)) and post-SD (right panels, 2 h during recovery after sleep onset). TPer2 KO mice do not sleep more than controls pre- as well as post-SD. Two-way ANOVA with Šídák's post-test, $n = 22$ *wt*, $n = 24$ T*Per2*; (**G**) brief awakenings in T*Per2* mice are comparable to control animals. Two-way ANOVA with Šídák's post-test. (**H**) average sleep bout duration pre-SD (2 h in BL from ZT13 to ZT15, hours 13–14 in (**B**)) and post-SD (2 h during recovery from ZT13 to ZT15, hours 38–39 in (**B**)). Sleep bout duration is significantly decreased in T*Per2* KO mice postTwoSD. 2-way ANOVA with Šídák's post-test with ** $p < 0.01$, *** $p < 0.001$, $n = 22$ *wt*, $n = 24$ T*Per2*; (**I**) sleep bout durations pre-SD (left panels, 2 h in BL from ZT13 to ZT15, hours 13–14 in (**B**)) and post-SD (right panels, 2 h during recovery from ZT13 to ZT15, hours 38–39 in (**B**)). T*Per2* mice had more short sleep bouts in pre-SD as well as post-SD, *** $p < 0.001$, **** $p < 0.0001$, but wild-type controls showed a higher sleep percentage in mid and long sleep bouts in pre-SD as well as post-SD. Two-way ANOVA with Šídák's post-test with * $p < 0.05$, ** $p < 0.01$, *** $p < 0.001$, $n = 22$ *wt*, $n = 24$ T*Per2*; (**J**) brief awakenings are comparable to controls in T*Per2* mice pre-SD and post-SD. Two-way ANOVA with Šídák's post-test, $n = 22$ *wt*, $n = 24$ T*Per2*.

To evaluate the state of the sleep homeostat in the early dark phase, sleep parameters were determined for a 2-h time window from ZT13 to ZT15 before and after SD (blue squares in Figure 2B, hours 13–14 pre-SD, hours 38 and 39 post-SD). Sleep bout lengths tended to be shorter in T*Per2* mice pre-SD and were significantly shorter post-SD since controls reacted to SD with a lengthening (Figure 2H). Total sleep amount followed the same pattern with less total sleep in T*Per2* mice after SD (Figure S4B). In contrast to the light phase (Figure 2F), the number of short sleep bouts in the dark phase were significantly heightened in T*Per2* mice pre- as well as post-SD (Figure 2I, upper panels). This was accompanied by an increased sleep percentage spent in the medium-short sleep bout range and a decreased sleep percentage in the medium-long sleep bout range under both conditions (Figure 2I, lower panels). Short sleep bouts from pre- to post-SD were considerably reduced only in controls (Figure S5B). Analogous to the light phase, sleep was not more or less disrupted, seeing as brief awakenings in the dark phase pre- and post-SD were similar in T*Per2* mice compared to controls (Figure 2J).

Taken together, T*Per2* mice have an earlier sleep onset compared to controls after SD just like the *Per2*^*Brdm* mice. Furthermore, T*Per2* mice gain less sleep and have a more fragmented sleep pattern in the dark/active period (ZT13-15), something we also found in the *Per2*^*Brdm* mice. In contrast to the *Per2*^*Brdm* animals, T*Per2* mice did not display an altered sleep–wake pattern during their active phase. Likewise, their sleep was not more consolidated immediately after SD. Thus, in T*Per2* mice only, the faster sleep onset response points to a higher accumulated sleep pressure during SD due to a lack of *Per2*.

### 2.3. NPer2 Mice Display Less Fragmented Sleep under Undisturbed Conditions

Mice with a deletion of *Per2* in neuronal cells [25], designated as N*Per2* KO mice, were investigated for sleep in the PiezoSleep system (Figure 3). Data were collected as described above under 2.1. No significant differences in hourly amount of sleep between N*Per2* KO mice and their controls are observable throughout the baseline time-course (Figure 3A,B). The sleep profile during the light phase immediately after SD was also similar between the two genotypes with only controls gaining significantly more sleep compared to baseline at

hour 45 (=ZT21). The sleep profile at the end of the recovery dark phase is significantly different between genotypes and similar to the end of the dark period in baseline. During the baseline dark phase, N*Per2* KO mice gained significantly more total sleep than controls, while activity remained similar (Figure S1). Sleep bout distribution analysis in the light and dark phase revealed that N*Per2* KO mice showed less short sleep bouts than controls in the light (>0.5 min) and dark phase (>1 and >2 min) (Figure 3C, upper panels). The distribution of sleep percentage in the light phase was similar between the two genotypes whereas in the dark phase sleep percentage in N*Per2* KO animals was significantly decreased in shorter bouts (>2 min) and increased in longer bouts (>8 and >16 min, Figure 3C, lower panels). After SD, sleep onset was comparable between N*Per2* KO mice and controls (Figure 3D).

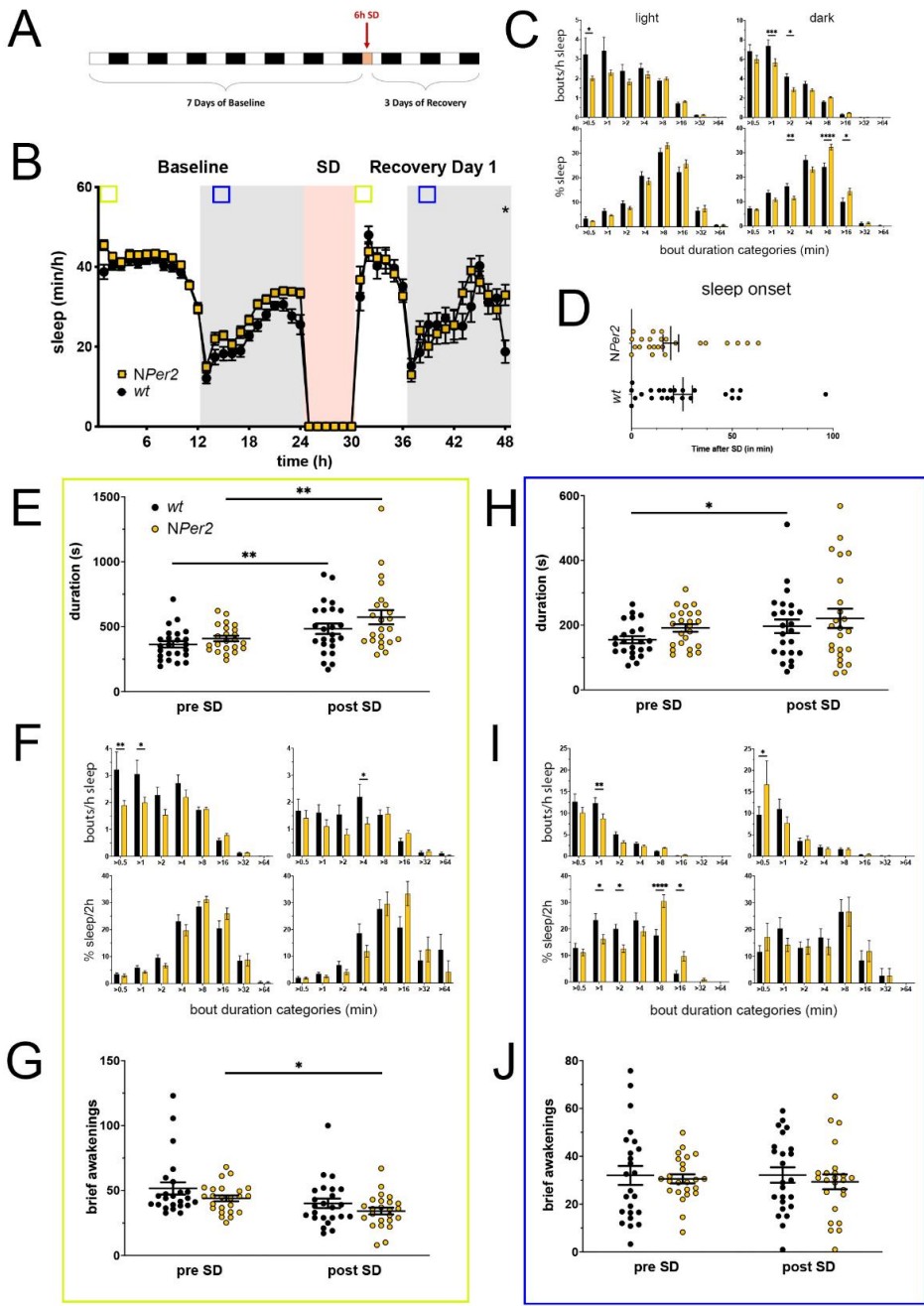

**Figure 3.** Sleep–wake analysis of neuronal *Per2* knock-out (N*Per2* KO) mice (yellow) compared to wild type controls (black). (**A**) Scheme of sleep–wake recordings during undisturbed conditions

(baseline = BL), sleep deprivation (SD), and sleep–wake recordings during recovery from SD; (**B**) time course of mean (±SEM) hourly values of sleep over 7 days summarized over 24 h baseline. White areas depict 12 h of light and gray shaded areas 12 h of darkness. The rose shaded area depicts 6 h of sleep deprivation (SD) in the light portion. The green and blue squares depict pre- and post-SD time windows being analyzed in detail in E-G (green square, 2 h light phase) and H-J (blue square, 2 h dark phase). Statistically significant differences of sleep between the two genotypes by two-way ANOVA with Šídák's post-test is depicted with an asterisk (* $p < 0.05$, $n = 24$ *wt*, $n = 24$ N*Per2*); (**C**) distribution of sleep bout durations in the light (left panels) and dark phase (right panels) of baseline sleep (hours 0–24 in (**B**)). The top panels show the number of bouts per hour of sleep in each sleep bout category in the light and dark phase. The bottom panels show the percentage of sleep in each of the categories. Values are shown as mean (±SEM). Two-way ANOVA with Šídák's post-test revealed statistically significant differences between the two genotypes with * $p < 0.05$, ** $p < 0.01$, *** $p < 0.001$, **** $p < 0.0001$, $n = 24$ *wt*, $n = 24$ N*Per2*; (**D**) sleep onset after SD. The N*Per2* KO mice show a similar onset compared to wild type controls. Mann–Whitney test, $n = 24$ *wt*, $n = 24$ N*Per2*; (**E**) average sleep bout duration pre-SD (2 h in BL from ZT0 to ZT2, hours 1–2 in (**B**)) and post-SD (2 h during recovery after sleep onset). Sleep bout duration is comparable between N*Per2* KO and control mice. Two-way ANOVA with Šídák's post-test, $n = 24$ *wt*, $n = 23$ N*Per2*; (**F**) sleep bout durations pre-SD (left panels, 2 h in BL from ZT0 to ZT2, hours 1–2 in (**B**)) and post-SD (right panels, 2 h during recovery after sleep onset). N*Per2* mice have less short bouts pre-SD and less medium bouts post-SD compared to controls. Two-way ANOVA with Šídák's post-test with * $p < 0.05$, ** $p < 0.01$, $n = 24$ *wt*, $n = 24$ N*Per2*; (**G**) brief awakenings are comparable between N*Per2* mice and controls. Two-way ANOVA with Šídák's post-test, * $p < 0.05$, $n = 24$ *wt*, $n = 24$ N*Per2*; (**H**) average sleep bout duration pre-SD (2 h in BL from ZT13 to ZT15, hours 13–14 in (**B**)) and post-SD (2 h during recovery from ZT13 to ZT15, hours 38–39 in (**B**)). Sleep bout duration is comparable between N*Per2* KO mice and controls. Two-way ANOVA with Šídák's post-test, * $p < 0.05$, $n = 24$ *wt*, $n = 24$ N*Per2*; (**I**) sleep bout durations pre-SD (left panels, 2 h in BL from ZT13 to ZT15, hours 13–14 in (**B**)) and post-SD (right panels, 2 h during recovery from ZT13 to ZT15, hours 38–39 in (**B**)). N*Per2* mice had less short sleep bouts in pre-SD but more in post-SD. N*Per2* mice showed a higher sleep percentage in mid and long sleep bouts in pre-SD, but not post-SD Two-way ANOVA with Šídák's post-test with * $p < 0.05$, ** $p < 0.001$, **** $p < 0.0001$, $n = 24$ *wt*, $n = 24$ N*Per2*; (**J**) brief awakenings are comparable between N*Per2* and control mice. Two-way ANOVA with Šídák's post-test, $n = 24$ *wt*, $n = 24$ N*Per2*.

Sleep bout duration in the light phase during the two hours of highest sleep pressure was comparable between N*Per2* KO mice and controls pre- as well as post-SD; both genotypes reacted to SD with a prolongation of their average sleep bout length (Figure 3E). Total sleep within this 2-h window was already higher in N*Per2* KO mice, but increased in controls to similar levels post-SD (Figure S2C). Furthermore, the number of short sleep bouts was decreased in N*Per2* mice pre-SD (>0.5 and >1 min) as well as post-SD (>4 min) (Figure 3F, upper panels). This distribution did not result in differences of time spent sleeping in sleep bout length bins between genotypes (Figure 3F, lower panels). Brief awakenings were not different between N*per2* KO and control animals and were overall reduced post-SD, indicating functional wake suppression (Figure 3G). Overall, N*Per2* KO mice gave a similar recovery response in this time window after sleep homeostasis was challenged with SD, but already exhibited signs of higher sleep consolidation under undisturbed conditions.

During the 2 h from ZT13 to ZT15 in the dark phase (blue squares in Figure 3B), the increase of average sleep bout length after SD was only significant for controls, but sleep bout duration remained similar for N*Per2* and control mice, pre-SD (in BL, hours 13–14 in Figure 3B) as well as post-SD (during recovery, hours 38–39 in Figure 3B), although N*Per2* mice displayed, on average, slightly longer sleep bouts in both conditions (not significant) (Figure 3H). The same was the case for the amount of sleep in these two hours, with control mice increasing their sleep amount from pre-SD to post-SD and N*Per2* mice maintaining a (not significantly different) higher sleep amount from pre-SD to post-SD (Figure S4C). The number of short sleep bouts in the 2-h window during the dark phase was significantly

decreased in N*Per2* mice pre-SD (>1 min), accompanied by a decreased sleep percentage in the short sleep bout range (>1, >2 min) and an increased sleep percentage in the medium-long sleep bout range (>8, >16 min), whereas post-SD short bout numbers were increased (>0.5 min) compared to controls, but distribution of sleep percentages remained unaffected (Figure 3I). Comparable to the light phase, brief awakenings in the dark phase pre- and post-SD were similar in N*Per2* mice and controls (Figure 3J).

Taken together, it appears that the N*Per2* mice may have already less fragmented sleep during baseline conditions in the light (ZT0-2) and the dark period (ZT13-15), which is maintained as response after SD, something control animals only show post-SD as a recovery response.

### 2.4. GPer2 Mice Display Less Fragmented Sleep in the Light Phase

Mice with a deletion of *Per2* in astroglial cells [26], designated as G*Per2* KO mice, were investigated for sleep in the PiezoSleep system (Figure 4). Data were collected as described above under 2.1. There are no significant differences in hourly sleep between G*Per2* KO mice and their controls (Figure 4A,B). Like the N*Per2* KO mice, G*Per2* KO mice gained a miniscule, but significant amount of more total sleep during the dark phase in baseline than controls, while activity remained similar (Figure S1).

The sleep profile after SD was also similar between the two genotypes. Post-SD in the dark phase after the initial wake peak at dark onset, controls slept significantly more than during baseline (hours 39, 40, and 44), visible by the more pronounced rise in sleep amount. Sleep structure in the light and dark phase of baseline was similar between G*Per2* KO mice and their controls: the number of sleep bouts as well as sleep percentage were distributed identically (Figure 4C). After SD, sleep onset was comparable between G*Per2* KO mice and controls (Figure 4D).

Sleep bout duration within the 2 h of highest sleep pressure in the light phase was comparable between G*Per2* KO mice and controls pre- as well as post-SD, with both increasing from pre-SD to post-SD (Figure 4E). The same was observed for total sleep (Figure S2D). However, the number of very short sleep bouts were decreased in G*Per2* mice pre-SD (>0.5 min), but not post-SD (Figure 4F, upper panels)—similar to the N*Per2* KO mice. This did not impact the distribution of time spent asleep in the different sleep bout length bins pre-SD and post-SD (Figure 4F, lower panels). Both genotypes reacted to SD with a decrease of short sleep bout numbers and an increase of time spent asleep in longer sleep bouts ). Brief awakenings were not different between G*Per2* KO and control animals (Figure 4G).Sleep bout duration within the 2 h of highest sleep pressure in the light phase was comparable between G*Per2* KO mice and controls pre- as well as post-SD, with both increasing from pre-SD to post-SD (Figure 4E). The same was observed for total sleep (Figure S2D). However, the number of very short sleep bouts were decreased in G*Per2* mice pre-SD (>0.5 min), but not post-SD (Figure 4F, upper panels)—similar to the N*Per2* KO mice. This did not impact the distribution of time spent asleep in the different sleep bout length bins pre-SD and post-SD (Figure 4F, lower panels). Both genotypes reacted to SD with a decrease of short sleep bout numbers and an increase of time spent asleep in longer sleep bouts ). Brief awakenings were not different between G*Per2* KO and control animals (Figure 4G).

Taken together, our data indicate that astroglial *Per2* has a minor influence on sleep composition in the early light phase by altering the number of very short sleep bouts, but no impact on sleep homeostasis was distinguishable.

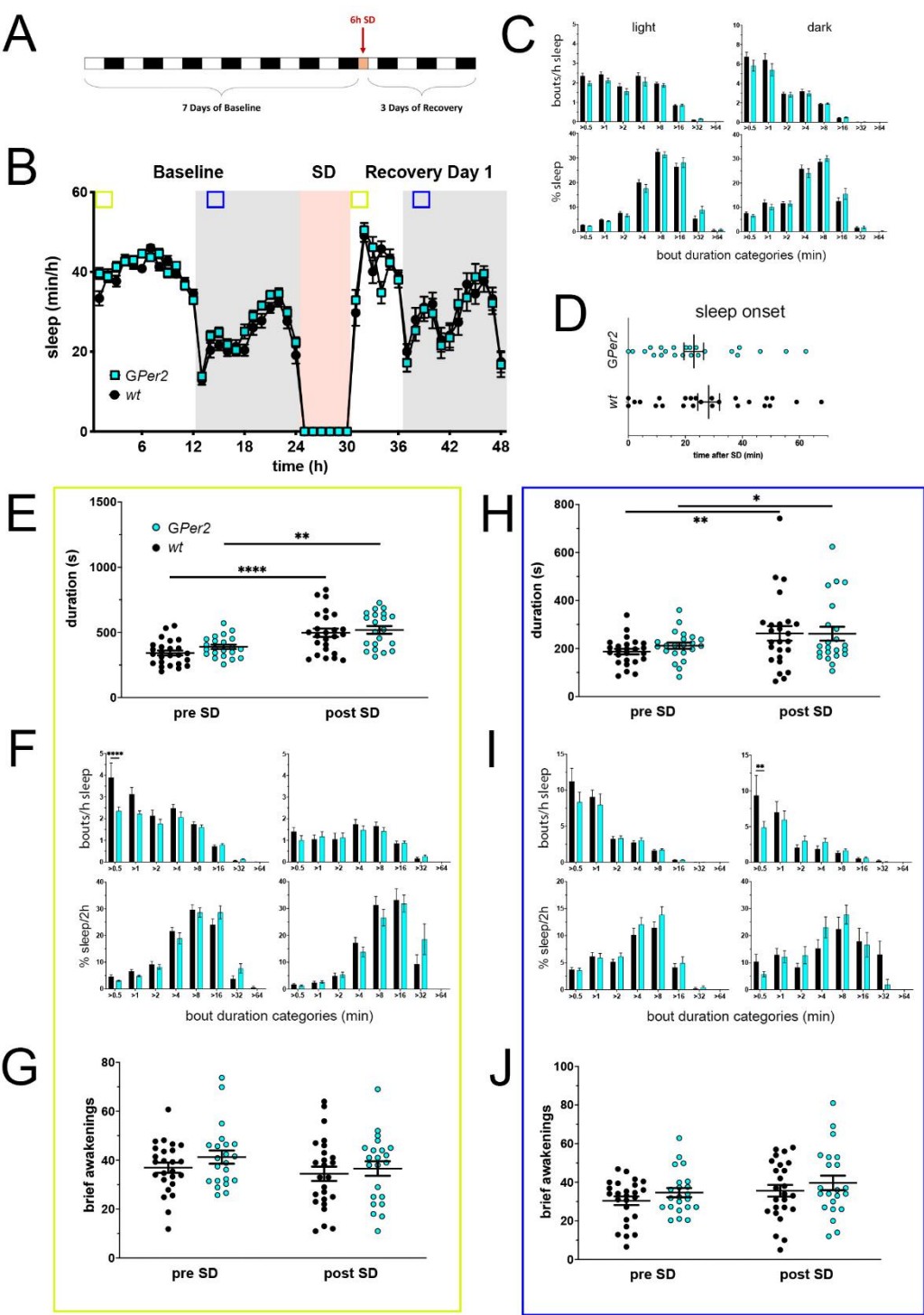

**Figure 4.** Sleep–wake analysis of glial *Per2* knock-out (*GPer2* KO) mice (blue) compared to wild type controls (black). (**A**) Scheme of sleep–wake recordings during undisturbed conditions (baseline = BL), sleep deprivation (SD), and sleep–wake recordings during recovery from SD; (**B**) time course of mean (±SEM) hourly values of sleep over 7 days summarized over 24 h baseline. White areas depict 12 h of light and gray shaded areas 12 h of darkness. The rose shaded area depicts 6 h of sleep deprivation (SD) in the light portion. The green and blue squares depict pre- and post-SD time windows being analyzed in detail in E-G (green square, 2 h light phase) and H-J (blue square, 2 h dark phase). There is no statistically significant difference of sleep between the two genotypes. Two-way ANOVA with Šídák's post-test (*n* = 25 *wt*, *n* = 22 *GPer2*); (**C**) distribution of sleep bout durations in the light (left panels) and dark phase (right panels) of baseline sleep (hours 0–24 in (**B**)).

The top panels show the number of bouts per hour of sleep in each sleep bout category in the light and dark phase. The bottom panels show the percentage of sleep in each of the categories. Values are shown as mean (±SEM). Two-way ANOVA with Šídák's post-test revealed no statistically significant differences between the two genotypes, *n* = 25 *wt*, *n* = 22 G*Per2*; (**D**) sleep onset after SD. The G*Per2* KO mice show a similar onset compared to wild type controls. Mann–Whitney test with *n* = 25 *wt*, *n* = 23 G*Per2*; (**E**) average sleep bout duration pre-SD (2 h in BL from ZT0 to ZT2, hours 1–2 in (**B**)) and post-SD (2 h during recovery after sleep onset). Sleep bout duration is comparable between G*Per2* KO and control mice. Two-way ANOVA with Šídák's post-test, ** *p* < 0.01, **** *p* < 0.0001, *n* = 25 *wt*, *n* = 22 T*Per2*; (**F**) sleep bout durations pre-SD (left panels, 2 h in BL from ZT0 to ZT2, hours 1–2 in (**B**)) and post-SD (right panels, 2 h during recovery after sleep onset). G*Per2* KO mice show less short bouts pre-SD, but have an increased sleep percentage in longer bouts pre- and post-SD compared to controls. Two-way ANOVA with Šídák's post-test, *n* = 25 *wt*, *n* = 22 G*Per2*; (**G**) brief awakenings in G*Per2* mice are comparable to control animals. Two-way ANOVA with Šídák's post-test (*n* = 25 *wt*, *n* = 22 G*Per2*); (**H**) average sleep bout duration pre-SD (2 h in BL from ZT13 to ZT15, hours 13–14 in (**B**)) and post-SD (2 h during recovery from ZT13 to ZT15, hours 38–39 in (**B**)). Sleep bout duration is comparable between G*Per2* KO mice and controls. Two-way ANOVA with Šídák's post-test, * *p* < 0.05, ** *p* < 0.01, *n* = 25 *wt*, *n* = 22 G*Per2*; (**I**) sleep bout durations pre-SD (left panels, 2 h in BL from ZT13 to ZT15, hours 13–14 in (**B**)) and post-SD (right panels, 2 h during recovery from ZT13 to ZT15, hours 38–39 in (**B**)). G*Per2* mice had less short sleep bouts post-SD. Two-way ANOVA with Šídák's post-test with ** *p* < 0.01, *n* = 25 *wt*, *n* = 22 G*Per2*; (**J**) brief awakenings are comparable to controls in G*Per2* mice pre-SD and post-SD. Two-way ANOVA with Šídák's post-test, *n* = 25 *wt*, *n* = 22 G*Per2*.

In the 2 h of the dark phase (ZT13 to ZT15, blue squares in Figure 4B), sleep bout duration and total sleep were similar in G*Per2* and control mice, pre-SD (hours 13–14 in Figure 4B) as well as post-SD (hours 38–39 in Figure 4B) (Figure 4H and Figure S4D). The number of short sleep bouts in this time window were significantly decreased in G*Per2* mice post-SD (>0.5 min), but not pre-SD compared to controls (Figure 4I, upper panels). The distribution of sleep percentages per bout duration category pre- as well as post-SD were comparable between the two genotypes (Figure 4I, lower panels), with an expected drop of short sleep bouts and an increase of time spent in longer sleep bouts (Figure S5D). Similar to the light phase, brief awakenings in the dark phase pre- and post-SD did not differ between G*Per2* mice and controls (Figure 4J).

Taken together, G*Per2* mice may have slightly more consolidated sleep in the light/resting period (ZT0-2) during baseline conditions, but no effect of lacking *Per2* in astroglia on the homeostatic sleep response was discernable.

### 2.5. Whole Body Per2-Deficient Mice have Similar Sleep but Different Metabolic Parameters in Metabolic Cages

Some of the phenotypes observed for the *Per2^Brdm1^* and T*Per2* mice may be due to underlying differences in their metabolism as compared to their controls. Consequently, we set out to determine the link between the sleep phenotype and metabolic parameters for these mice using a full mouse phenotyping system (Sable System). Male 4- to 5-month-old T*Per2* and control mice did not show significant differences in body weight, fat mass (FM), or fat-free mass (FFM) (Table 1). Meanwhile, the *Per2^Brdm^* were significantly heavier than the controls, but this was accounted for solely by an increase in fat-free mass (FFM). This difference in body weight and composition pushed us to analyze normalized (i.e., related to body composition) energy expenditure, food intake, and water intake measurements (Table 1). Sleep percentage (Figure 5A,B) confirmed what had been observed in Sections 2.1 and 2.2 (see Figures 1 and 2). The *Per2^Brdm^* and T*Per2* mice increased activity at the end of the lights on period (~ZT 9/10) that correlated with significant increases in food and water intake in the *Per2^Brdm^* (Table 1) mice only. Total energy expenditure normalized to FFM showed no gross differences between *Per2^Brdm^* (Figure 5C) and T*Per2*

(Figure 5D) with their respective controls and seemed to logically follow sleep patterns. The average 24 h respiratory quotient (RQ) was not significantly different amongst the groups (Table 1), although if broken down into light and dark phases, an elevated RQ in the *Per2*[Brdm] (Figure 5E) corresponded to the increase in food intake observed (rising RQ represents a switch to more carbohydrate oxidation) reflecting increased feeding behavior (Table 1). A similar rise in RQ was observed in the T*Per2* (Figure 5F) mice at the end of the lights on period (~ZT 8/9). Nonetheless, the overall RQ of T*Per2* in the light phase was less than in controls (Table 1). No differences in ambulatory activity were observed between the T*Per2* mice and their controls, but lower total and dark phase activity was observed in the *Per2*[Brdm]. To highlight the daytime-dependent differences in food intake, water consumption, and ambulatory activity between the mouse strains, the cumulative values are displayed in Figure S6. Taken together, the data would indicate that both types of *Per2*-deficient mice displayed similar sleeping behaviors in the metabolic cages, but had different metabolic parameters over time. Hence, there is no obvious correlation between the sleep phenotype and their metabolism, except for the number of brief awakenings pre-SD (Figures 1 and 2).

**Table 1.** Metabolic Phenotyping. Total energy expenditure (TEE), food intake, and water intake are normalized to FFM. Total ambulatory activity represents all directed ambulatory locomotion (at least 1 cm/s) within the beam break system. *t*-tests were used to compare knock-out animals to their respective controls; values are means ±SEM ** $p < 0.01$, * $p < 0.05$, $n = 12$ per genotype.

| | Wt | *Per2*[Brdm] | Wt (fl/fl) | T*Per2* (D/D) |
|---|---|---|---|---|
| Body weight (g) | 23.7 ± 0.7 ** | 34.9 ± 1.1 | 31.9 ± 0.7 | 31.7 ± 0.7 |
| Fat mass (FM—g) | 2.4 ± 0.8 | 2.8 ± 0.3 | 4.3 ± 0.7 | 5.7 ± 0.5 |
| Fat free mass (FFM—g) | 20.3 ± 0.5 ** | 30.2 ± 0.8 | 25.8 ± 0.8 | 24.3 ± 0.5 |
| Total energy expenditure (TEE/FFM)—kcal/24 h/g) | 0.47 ± 0.02 | 0.51 ± 0.02 | 0.43 ± 0.02 | 0.45 ± 0.01 |
| Respiratory Quotient (RQ) | 0.83 ± 0.01 | 0.84 ± 0.01 | 0.83 ± 0.01 | 0.81 ± 0.01 |
| Respiratory Quotient (RQ light) | 0.82 ± 0.02 ** | 0.85 ± 0.02 | 0.84 ± 0.01 * | 0.81 ± 0.01 |
| Respiratory Quotient (RQ dark) | 0.84 ± 0.02 | 0.82 ± 0.01 | 0.82 ± 0.01 | 0.82 ± 0.01 |
| Food intake/FFM (Kcal/24 h/g) | 0.54 ± 0.04 | 0.51 ± 0.04 | 0.39 ± 0.04 | 0.44 ± 0.05 |
| Food intake (light phase—kcal/12 h) | 3.3 ± 0.7 ** | 7.4 ± 0.4 | 3.6 ± 0.6 | 3.3 ± 0.5 |
| Food intake (dark phase—kcal/12 h) | 7.6 ± 0.8 | 7.8 ± 0.9 | 6.2 ± 0.5 | 7.5 ± 1.1 |
| Water intake/FFM (g/24 h/g) | 0.16 ± 0.06 | 0.19 ± 0.04 | 0.13 ± 0.03 | 0.14 ± 0.02 |
| Water intake (light phase—g) | 0.6 ± 0.1 ** | 2.7 + 0.2 | 1.1 ± 0.2 | 1.2 ± 0.3 |
| Water intake (dark phase—g) | 2.7 ± 0.2 | 3.1 ± 0.3 | 2.2 ± 0.3 | 2.1 ± 0.3 |
| Total Ambulatory Activity (meters) | 123.5 ± 9.7 ** | 89.2 ± 8.3 | 105.1 ± 8.2 | 116.83 ± 9.7 |
| Ambulatory Activity Light (meters) | 38.2 ± 5.4 | 35.2 ± 4.1 | 43.8 ± 3.0 | 40.1 ± 4.9 |
| Ambulatory Activity Dark (meters) | 85.3 ± 7.5 ** | 54.1 ± 5.0 | 61.3 ± 6.8 | 76.7 ± 7.3 |

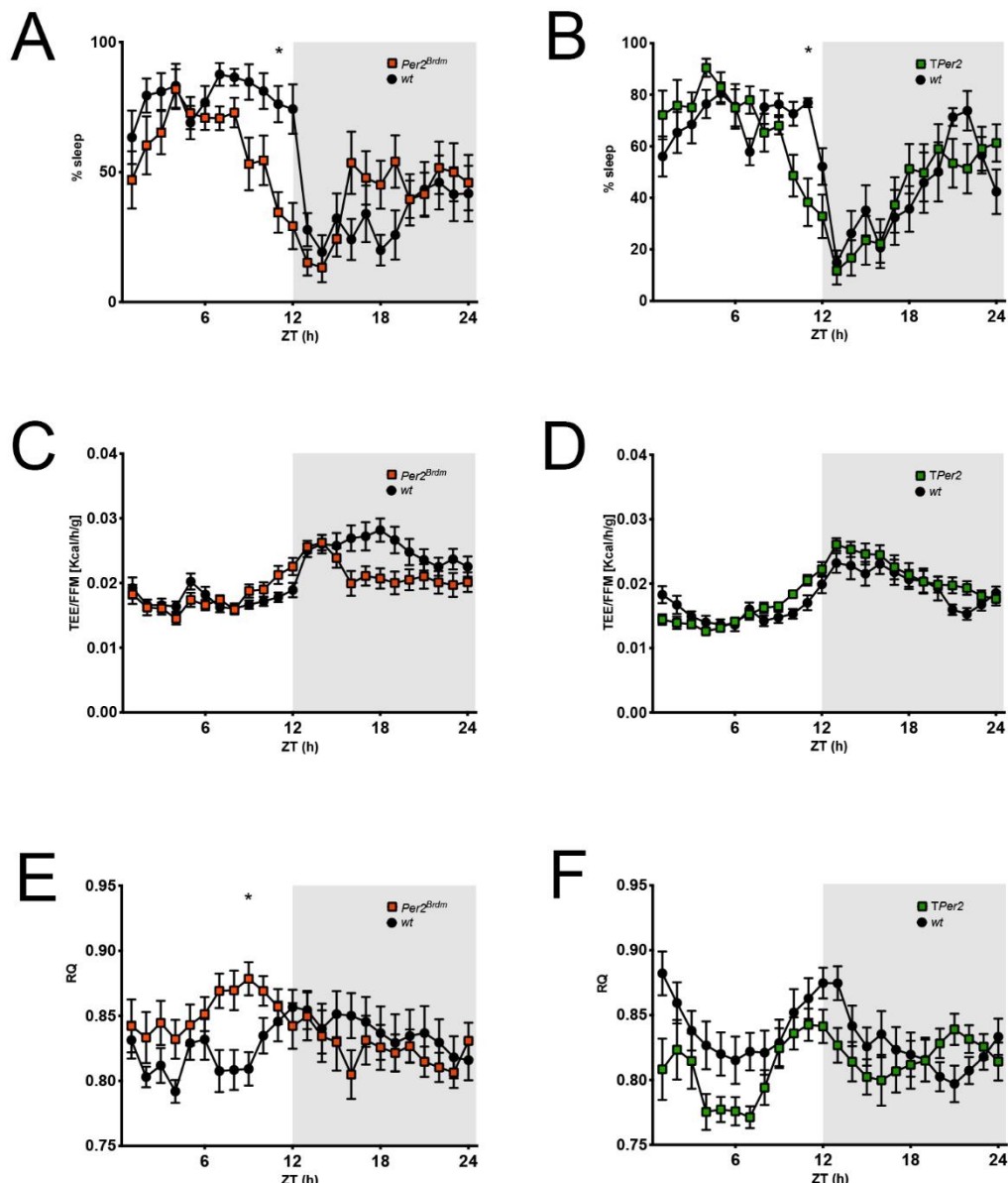

**Figure 5.** Comparison of sleep and metabolic data derived from the metabolic cages. (**A,B**) Percent of sleep per 1 h observation interval defined as absence of any movement sensor activation. *Per2^Brdm* and T*Per2* mice have an earlier wake onset than controls; (**C,D**) total energy expenditure (TEE) per 1 h normalized to the fat-free mass (FFM) of each mouse; (**E,F**) respiratory quotient (RQ) per 1 h defined as $CO_2$ produced/$O_2$ consumed. Corresponding to an earlier wake onset, RQ increases earlier in *Per2^Brdm* and T*Per2* mice. For all panels: $n = 12$, repeated measures two-way ANOVA with Šídák's correction for multiple testing, * $p < 0.05$.

## 3. Discussion

In this study we investigated the influence of lack of the *Per2* gene specifically in neurons (N*Per2*), astroglial cells (G*Per2*), or in all body cells (T*Per2* and *Per2^Brdm*) on sleep. Since T*Per2* and *Per2^Brdm* showed a sleep phenotype, only these two genotypes were assessed for metabolic parameters. We found that lack or mutation of *Per2* in all body cells (T*Per2* and *Per2^Brdm* mice, respectively) led to early onset activity at the light/dark transition (Figures 1B, 2B and 5A,B). After SD, sleep onset was significantly faster in both genotypes compared to control animals (Figures 1D and 2D). A total of 8–9 h after SD (recovery dark phase), sleep duration was significantly shorter in both genotypes and appeared to be more fragmented (Figure 1H,I and Figure 2H,I). Both genotypes maintained

their fragmented sleep pattern they exhibit already under baseline conditions and they did not display an increase in sleep consolidation or sleep depth. These observations are partially consistent with previous reports [7,23]. Earlier sleep onset after SD and a less consolidated homeostatic recovery response in the early night after SD point towards a role of *Per2* in the accumulation of sleep pressure during enforced wakefulness and in the subsequent homeostatic reaction in the early dark period to return to the sleep–wake equilibrium. *Per2* seems to be aiding the organism in coping with the mounting need to sleep during prolonged waking and to be involved in the physiological rebound response during the early night.

However, we also observed differences between the two whole body cell knock-out strains T*Per2* and *Per2*^*Brdm*. The *Per2*^*Brdm* animals showed a stronger recovery response to SD than controls in the first 2 h immediately after SD in the light phase (Figure 1E,F), which was not observed in T*Per2* mice (Figure 2E,F). While the number of brief awakenings under baseline conditions before SD were increased in *Per2*^*Brdm* animals, the brief awakenings in T*Per2* mice were comparable to controls Figures 1G and 2G). *Per2*^*Brdm* mice are on a mixed SV129/B6 background, T*Per2* on a B6, which could explain the differences between both strains. An earlier study assessing sleep in six inbred mouse strains, including SV129 and B6, reported differences and attributed those to genetic background [30]. This also explains the variations between the different wild-types used. Still, differences between both total knock-outs prompted us to investigate whether the two strains were metabolically different (Figure 5). We found that the respiratory quotient, a measure for the utilization of carbohydrates (RQ of 1.0) or fat (RQ of 0.7), was significantly different between the two strains. The *Per2*^*Brdm* mice had in general a higher RQ than T*Per2* animals in the light phase (Figure 5E,F), which is consistent with an increased intermittent light-phase feeding behavior that correlates well with the increased number of brief awakenings observed in *Per2*^*Brdm* mice (Figure 1G). Hence, this parameter is probably influenced by metabolism and feeding behavior. The recovery response to SD is probably affected by metabolism as well. This interpretation is supported by a previous study showing that SD is associated with food deprivation and the resulting energy deficit may contribute to the effects observed after SD that were commonly interpreted as a response to sleep loss only [31].

Deletion of *Per2* in specific brain cells only only marginally affected sleep parameters under undisturbed conditions and after sleep deprivation. Nonetheless, N*Per2* and G*Per2* KO mice gained, on average, 48 min and 36 min, respectively, more sleep in the dark phase under unchallenged conditions. Because locomotion was unchanged, the increase in sleep amount was not due to diminished or altered physical activity. We deduce from this that either neuronal or astrocytic *Per2* contribute on a minor scale to overall sleep need during the active phase in the state of sleep–wake equilibrium. However, when the sleep–wake ratio set-point is thrown off balance, both neuronal and astrocytic *Per2* seem not to be involved in the homeostatic recovery response. Additionally, a minor change throughout all conditions (light/dark, pre-/post-SD) was observable in sleep bout composition with less short sleep bouts and more time spent sleeping in longer sleep bouts. This suggests, in general, more consolidated sleep when *Per2* is lacking in either neurons or astrocytes and is in line with the minuscule increase in total sleep. Nevertheless, overall, neither deletion of *Per2* in neurons (Figure 3) nor in astroglial cells (Figure 4) affected sleep parameters compared to controls. This observation can be interpreted in several ways. It appears that the deletion of *Per2* in specific brain cells does not affect sleep, although the metabolic communication between neurons and astroglia was probably not normal. However, this point would have to be verified. Furthermore, the results suggest that impairing the clock in specific cell populations of the brain has hardly any effect on sleep parameters, probably due to the presence of normal functioning clocks in the periphery. This would suggest that sleep is a systemic phenomenon that is not wholly dependent on the brain only and that peripheral clocks may be involved in sleep regulation. In support of this view is a study which showed that the clock gene *Bmal1* in skeletal muscle regulates sleep [32]. Restoration of *Bmal1* expression in the brain of *Bmal1* whole body knock-out (*Bmal1* KO)

animals did not restore the sleep phenotype of the *Bmal1* KO strain, while restoring the expression of this gene in skeletal-muscle did [32]. Therefore, it would be interesting to test whether deletion of *Per2* in peripheral organs, such as the muscle only, would be sufficient to affect sleep. However, another interpretation of our data could be that the sleep phenotypes observed for the whole-body knock-out mice (*Per2^{Brdm}* and T*Per2*) stem from the simultaneous knock-out of both neurons and glia. Compensatory mechanisms of either neurons or astroglial cells might be sufficient to rescue the sleep regulation response of the cell-specific knock-outs. Therefore, it remains to be seen whether a lack of *Per2* in neurons and astrocytes combined or *Per2* in peripheral organs contributes to the sleep phenotype observed in whole body *Per2* knock-out mice.

A limitation of our study is that sleep was measured non-invasively and no electroencephalographic (EEG) data was collected. Therefore, no distinction between different sleep stages (non-rapid eye movement/rapid eye movement) were made. Due to the lack of EEG data, we were not able to measure delta power, a parameter believed to index homeostatic sleep. However, recent data indicate that delta power is not as representative of prior sleep–wake history as previously thought [33]. Nonetheless, not being able to differentiate sleep into REM and NREM potentially blinds to existing differences in sleep composition. As such, and in line with peripheral organs affecting sleep, it has been shown that liver Acyl-CoA thioesterase 11 (ACOT11), regulating free fatty acids (FFAs) through the mitochondrial beta-oxidation pathway, is involved in NREM sleep recovery [34].

Taken together, our study suggests that the circadian clock gene *Per2* impacts on aspects of sleep and sleep regulation, specifically altering the homeostatic sleep response by delaying the timing of sleep onset after enforced wakefulness. Since this sleep phenotype can neither be attributed exclusively to neuronal nor astroglial *Per2*, a deletion of *Per2* in all cells of the central nervous system and/or peripheral cells (e.g., muscle cells) may be important for sleep regulation.

## 4. Materials and Methods

### 4.1. Animals and Housing

Data from 182 mice contributed to this study. From 190 animals, 7 were excluded due to technical and 1 due to experimental failure. Male and female mice of 3–4 months of age were used. Animals were individually housed in polycarbonate cages (18 × 18 cm) in a 12 h light/12 h dark (12:12 LD) cycle. Food and water were available ad libitum and bedding, nestlet squares, and tissue were provided. Timing of experiments is expressed as zeitgeber time (ZT, ZT0 lights on, ZT12 lights off). Housing and experimental procedures were performed in accordance with the guidelines of the Schweizer Tierschutzgesetz and the Declaration of Helsinki. The state veterinarian of the Canton of Fribourg and the cantonal commission for animal experiments approved the protocol on 30.11.2018 (2018_36_FR, 30811).

Four different genetically modified mouse strains were used: *Per2^{Brdm}*, T*Per2*, N*Per2*, and G*Per2* knock-out mice. The generation of these strains have been described in detail elsewhere. Briefly, *Per2^{Brdm}* mice express a mutant PER2 protein due to an in-frame deletion, yielding an unstable protein that cannot translocate into the nucleus any longer and exert its function [27,29]. The *Per2^{Brdm}* mutant mice and their wild-type controls were on a mixed SV129 and C57BL/6 background. T*Per2*, N*Per2*, and G*Per2* knock-out mice were generated by crossing *Per2* floxed (*Per2*fl/fl) mice (European Mouse Mutant Archive (EMMA), strain EM:10599, B6JRj.129P2(129S3)-Per2tm1.1Ual/Biat) maintained on a C57BL/6 background [25] with mice carrying the *Cre recombinase* transgene under the control of specific promotors. *Per2*fl/fl mice have exon 6 of *Per2* flanked by two loxP sites. With the expression of Cre recombinase, exon 6 is deleted and exon 7 put out of frame leading to a truncated PER2 protein that is unstable and non-functional. T*Per2* KO mice with a complete deletion of *Per2* in all body cells were generated by breeding *Per2*fl/fl mice with mice that carry a *Cre recombinase* transgene under the control of the cytomegalovirus promotor (EMMA, strain EM:01149, B6.129-Cre-Deleter: B6.C(129)-Tg(CMV-cre)1Cgn/CgnIbcm, received from the Mouse Genetics Cologne Stiftung) [25,35]. N*Per2* mice with a deletion of *Per2* in nestin

positive neuronal were generated by breeding *Per2*fl/fl mice with mice carrying a *Cre recombinase* transgene under the control of the nestin promoter (EMMA, strain EM:04561, Bclaf1 × Tg Nes-cre C57BL/6: Tg(Nes-cre)1Kln9, received from the FMP Leibnitz-Institute für Molekulare Pharmakologie) [36]. To create G*Per2* KO mice with a deletion of *Per2* in astroglial cells [26], *Per2*fl/fl mice and the glial fibrillary acidic protein (GFAP)-driven Cre line (Jackson Lab, stock no. 004600, FVB-Tg(GFAP-cre)25Mes/J) were crossed with each other [37]. For N*per2* and G*per2* KO mice, genotyping after Cre-mediated recombination was performed by PCR. For N*per2* KO mice, 5′ GAG CAC TAG AGA AGG GAG TG 3′ and 5′ TCT GCA ATG TTG CCT CCC TG 3′ primers were used; for G *Per2* KO mice, 5′ ACT CCT TCA TAA AGC CCT 3′ and 5′ ATC ACT CGT TGC ATC GAC CG 3′ primers were used. T*Per2*, N*per2*, and G*Per2* knock-out mice were maintained on a C57BL/6 background. For T*Per2* KO mice, corresponding controls were *Per2*fl/fl mice as T*Per2* KO mice were kept as an individual strain with no remaining *Cre recombinase* after one-time knock-out. Control mice for N*Per2* and G*Per2* KO mice were littermates carrying either a single copy of the *Cre* transgene and being homozygous for the wild-type *Per2* allele or being heterozygous for *Per2* with one wild-type and one floxed *Per2* allele to control for any potential phenotype caused by *Cre* expression or LoxP sites.

### 4.2. Experimental Protocol (Sleep-Related)

Mice were placed in the piezo device (Signal Solutions, LLC, Lexington, KY, USA). After letting the mice habituate to their new environment for at least three days, they were left undisturbed to record baseline sleep–wake patterns for seven days. On the eighth day from ZT0 to ZT6, when sleep pressure was highest, mice were sleep-deprived for six hours. Sleep deprivation (SD) was achieved by gentle handling consisting of the introduction of novel objects, bedding changes, and interaction with the experimenter. After SD, recovery sleep was recorded for the subsequent three days.

### 4.3. Sleep Data Acquisition

Sleep–wake behavior was recorded using the non-invasive PiezoSleep system (Signal Solutions LLC, Lexington, KY, USA). A highly sensitive piezoelectric pad detecting pressure signals is placed underneath the cage separated only by a thin plastic sheet from the mouse. Mechanical pressure elicited by the animal's locomotor activity, heartbeat, and breathing is transformed into correlating electrical signals that are classified by the software's algorithm every two seconds as sleep ("1") or wake ("0") depending on the signal's strength and regularity. A typical sleep pattern consists of only very regular, but minuscule waves with an amplitude between 2 and 4 Hz due to the mere detected chest movement coming from the breathing of the sleeping mouse. Piezoelectric recordings do not allow for differentiation of sleep stages, but its classification accuracy of >90% to discriminate between sleep and wake has been validated with EEG [28,38,39].

### 4.4. Sleep Data Assessment

The quality of sleep recordings was controlled by examining the distribution of the decision statistic values obtained from the sleep–wake classifier. If the piezo decision algorithm failed to distinguish properly between the wake and sleep state and thus the typical bimodal distribution with either high ("sleep") or low ("wake") decision statistic values was not observable, the recording was excluded. A total of 6 out of 191 sleep recordings were excluded since they did not meet the necessary recording quality, likely due to too much bedding or nesting material interfering with breathing rate measurement.

### 4.5. Experimental Protocol (Metabolic Phenotyping)

*Per2*$^{Brdm}$, T*Per2*, and each group's respective controls (see under "Animals and housing" section above; *n* = 12 per group) were metabolically phenotyped using a 16-cage Promethion Core Metabolic System (Sable Systems International, North Las Vegas, NV, USA) maintained within an Environmental Control Cabinet with temperature maintained

at $26 \pm 0.2$ °C. Mice were between 4–5 months on average upon entering the system. Mice were individually housed and provided free access to water and food (normal chow) during a total period of 3 days. Mice were acclimatized for 48 h and the last 24 h were analyzed. $O_2$ consumption ($VO_2$) and $CO_2$ production ($VCO_2$) were used to calculate energy expenditure (expressed in either kcal/h or kcal/24 h) and respiratory quotient ($RQ = VCO_2/VO_2$). The RQ reflects which substrate is being oxidized by the animal with 1 representing pure carbohydrate oxidation and 0.7 pure lipid oxidation. Resting energy expenditure (REE in kilocalories per hour) was defined as the lowest 1-h period of energy expenditure for each individual mouse. Physical activity was measured by infrared beams integrated with the Promethion system with x, y, and z axes and feeding and drinking behavior were continuously measured during the entire test period. Sleep is defined by the system as having been "quiet" for more than 40 s. Animal's "quiet" time is defined as time in which the animal was not engaged in eating, drinking, grooming, nor locomotion and is represented as a percentage of 1 h bins spent sleeping. All mice had their body composition assessed at the beginning and at the end of the experimental period using an EchoMRI-100H body composition analyzer (EchoMRI LLC, Houston, TX, USA) and used to normalize energy expenditure data.

*4.6. Data Analysis*

Sleep related: Using a custom-made version from the developers, the accompanying software SleepStats was used to extract the binary files for each mouse. They contain every two-second decision over the whole length of the recording. With Matlab (Version R2019b, MathWorks Inc., Natick, MA, USA), the files were cropped to the desired length of 10 days starting at ZT0 of baseline day 1 and subsequent analyses were performed. Sleep minutes over specific time windows (per hour, per 12 h light, per 12 h dark, per 24 h) were computed. Hourly values were used to determine the time course of the amount of sleep and wake. Additionally, to calculating sleep quantity, sleep fragmentation as a measure of sleep quality was determined. For 12-h (light and dark period during baseline sleep) or 2-h (comparison of pre- and post-SD, in light and in dark) time frames and using a sliding window of 30-s (if it contained ≥50% sleep-positive decisions, it was counted as sleep) sleep episodes and their lengths were determined and their average duration calculated. Depending on their duration, sleep episodes were then sorted into one of eight duration categories of logarithmically increasing size (with lower bin limits of 0.5, 1, 2, 4, 8, 16, 32, and 64 min) and their frequency in each duration category expressed per hour of sleep by dividing the episode number by the total duration of sleep during the respective time frame. Normalization to hour of sleep is essential to partial out the potentially confounding variable of sleep amount if mice displayed a differential amount of sleep during the analyzed time frame. In addition to the frequency distribution, the amount of time the mice spent sleeping in each of the bout duration categories relative to the total time spent asleep in the analyzed time window was computed. Sleep onset after sleep deprivation was defined as the first sustained sleep episode longer than eight minutes. To determine sleep fragmentation, brief awakenings were defined as short wake bouts lasting ≤15 s and counted for 2-h time frames.

Metabolic Phenotyping: Energy expenditure, food intake, and water intake data in Table 1 are presented normalized by fat-free mass (FFM). As energy expenditure is proportional to body mass and composition (FFM) and fat mass (FM), a secondary set of more comprehensive analyses were conducted by creating multiple regression equations relating TEE and REE (kcal/24 h) to both FFM and FM (grams) using mice from the two control groups. This produced the two following equations:

$$TEE = 0.496 \times FFM - 0.310 \times FM; (p < 0.001 \text{ for FFM})$$

$$REE = 0.011 \times FFM + 0.005 \times FM; (p < 0.001 \text{ for FFM})$$

These equations were used to predict TEE and REE, respectively, for all animals, and residuals were calculated by subtracting predicted from observed values. *t*-test comparisons showed no significant differences between the residuals of KO and their respective control groups, further confirming the conclusion that there are no significant differences in adjusted metabolic rate amongst the groups. Data are expressed as group means $\pm$ SEM and statistical analyses were performed using JMP Pro 14 (SAS Institute, Cary, NC, USA). Statistical significance was prospectively defined as * P$\alpha$ < 0.05 and ** P$\alpha$ < 0.01. Student's *t*-tests and ANOVAs were used where applicable.

### 4.7. Wheel-Running Activity

Mice were individually housed in wheel-running cages in a 12 h light:12 h dark (LD) cycle, with free access to food and water. After a habituation period to stabilize activity, wheel running activity was recorded and analyzed using ClockLab (Actimetrics, Wilmette, IL, USA). The 7 days of activity were pooled and averaged for counts of wheel revolutions in the dark and the light phase.

### 4.8. Statistical Analysis

Statistics were calculated using GraphPad Prism software version 9.4.1. Depending on experimental design, Student's *t*-tests, ordinary or repeated measures two-way ANOVAs were performed. Significant effects were decomposed using post-hoc *t*-tests with Šídák correction for multiple comparisons or paired post-hoc *t*-tests. A *p*-value lower than 0.05 was considered statistically significant. All results are reported as mean $\pm$ standard error of the mean. Data from male and female mice were pooled in all analyses as no significant differences between sexes were observed.

**Supplementary Materials:** The following supporting information can be downloaded at: https://www.mdpi.com/article/10.3390/clockssleep5020017/s1, Figure S1: Sleep and activity in baseline during the light and the dark phase; Figure S2: Total time spent asleep in 2 h of the light phase before and after SD; Figure S3: Distribution of sleep bout durations comparing pre-SD and post-SD in 2 h of the light phase; Figure S4: Total time spent asleep in 2 h of the dark phase before and after SD; Figure S5: Distribution of sleep bout durations comparing pre-SD and post-SD in 2 h of the dark phase; Figure S6: Food intake, water consumption, and ambulatory activity over the course of a day.

**Author Contributions:** Conceptualization, K.S.W. and U.A.; methodology, K.S.W., J.A.R. and Y.R.; software, H.A. and K.S.W. validation, K.S.W., J.A.R. and Y.R.; formal analysis, K.S.W., J.A.R. and Y.R.; resources, G.R., U.A. and Y.R.; data curation, K.S.W. and U.A.; writing—original draft preparation, U.A.; writing—review and editing, K.S.W., J.A.R. and Y.R.; visualization, U.A.; supervision, U.A.; project administration, U.A.; funding acquisition, G.R. and U.A. All authors have read and agreed to the published version of the manuscript.

**Funding:** This research was funded by the Swiss National Science Foundation, grant number 3100_184667/1, a metabolic phenotyping grant 316030_205625, and the State of Fribourg.

**Institutional Review Board Statement:** Not applicable.

**Informed Consent Statement:** Not applicable.

**Data Availability Statement:** Not applicable.

**Acknowledgments:** We would like to thank Paul Franken from the University of Lausanne for invaluable advice and stimulating discussions as well as Antoinette Hayoz, Stéphanie Aebischer, and Maude Marmy for technical support. We would like to thank Isabelle Scerri for conducting the indirect calorimetry measurements.

**Conflicts of Interest:** The authors declare no conflict of interest.

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
