# Peer review of "Deletion of the Circadian Clock Gene Per2 in the Whole Body, but Not in Neurons or Astroglia, Affects Sleep in Response to Sleep Deprivation"

_2624-5175, doi:10.3390/clockssleep5020017_

Round 1

Reviewer 1 Report

The authors compared the effect of the lack of the Per2 gene in glial cells, neurons, and all body cells (TPer2 animals) in the mice sleep. The main results showed that the TPer2 animals displayed earlier sleep onset after sleep deprivation.

Author Response

We thank the reviewer for his time to read our manuscript.

Reviewer 2 Report

The authors have attempted to understand the function of per2 on sleep regulation by knocking out the same in the whole body and specifically on neuron as and structures. Mice with a mutation (Per2Brdm) or deletion in all cells of the body (TPer2) or specifically in neurons (NPer2) or glial cells (GPer2). By using a non-invasive PiezoSleep system, a sensitive pad which detects pressure. The authors conclude that Per2 contributes to the regulation of sleep onset after SD, in addition to balancing the normal number of sleep bouts and amount of sleep.

The study is well conducted and provides important insight into per2 function with  respect to neurons, astrocytes and whole body deletion. 

  1. The discussion is missing important information:  After SD, all mice went to sleep within 20 minutes after the end of SD, but Per2Brdm mice fell asleep significantly faster than controls  suggesting that they recovered sleep faster than controls- Discuss.

  1. It appears as if Per2 Brdm mice show more consolidated sleep after SD within this 2h window of highest sleep pressure- how 

  1. In Per2Brdm mice display in Fig. 1E, F, the number of short sleep bouts in the 2h slot within the dark phase were significantly increased in Per2Brdm mice pre- as well as post-SD- if this could be a compensatory mechanism? Discuss how Per2 contributes to this?

  1. Why there is no direct comparison graph between  Per2Brdm  and TPer2. Could be included.

  1. During the dark phase in baseline, NPer2 KO mice gained significantly more total sleep than controls while activity remained similar.  What does this signify discuss

  1.  How did the feeding behaviour compare with neuronal and astrocyte mice? Did they  change? 

  1. Since other glial cells (ex: microglia) are not per2 deleted. So  I recommend changing ‘glial’ to astroglial throughout the manuscript for clarity.

Author Response

The discussion is missing important information:  After SD, all mice went to sleep within 20 minutes after the end of SD, but Per2Brdm mice fell asleep significantly faster than controls suggesting that they recovered sleep faster than controls- Discuss.

See second paragraph of discussion.

It appears as if Per2 Brdm mice show more consolidated sleep after SD within this 2h window of highest sleep pressure- how 

Regarding this, we added to the results section:

“The immediate homeostatic response to SD with faster sleep onset and more consolidated sleep point to a higher accumulated sleep pressure during SD, something that would be attributable to the lack of Per2.”

However, we do not know how Per2 does that at the molecular level, which would of course be interesting to understand.

In Per2Brdm mice display in Fig. 1E, F, the number of short sleep bouts in the 2h slot within the dark phase were significantly increased in Per2Brdm mice pre- as well as post-SD- if this could be a compensatory mechanism? Discuss how Per2 contributes to this?

We assume, you are referring to Fig. 1H and I (2 hour window in the dark phase).

Sleep is already more fragmented in baseline conditions in Brdm mice. Therefore, we interpret their non-reaction after SD by not reducing their short sleep bout numbers as an attenuated sleep homeostatic response to SD. Hence, we assume that Per2 is implicated in this recovery process as written in the discussion. The exact mechanism by which Per2 exerts this, is still not known unfortunately.

Discussion part:

Earlier sleep onset after SD and a less consolidated homeostatic recovery response in the early night after SD point towards a role of Per2 in the accumulation of sleep pressure during enforced wakefulness and in the subsequent homeostatic reaction in the early dark period to return to the sleep-wake equilibrium. Per2 seems to be aiding the organism in coping with the mounting need to sleep during prolonged waking and to be involved in the physiological rebound response during the early night.

Why there is no direct comparison graph between Per2Brdm and TPer2. Could be included.

Per2Brdm and TPer2 have different genetic backgrounds, therefore we kept them separate. Also, it is easier like this to gain an overview over the different sleep parameters per strain. Additionally, four data points/time courses/distributions would render the figures very hard to understand and interpret.

In the supplemental figures though, all four genotypes are listed together per sleep parameter.

During the dark phase in baseline, NPer2 KO mice gained significantly more total sleep than controls while activity remained similar.  What does this signify discuss

We addressed this point in the discussion:

Although we did not observe changes in sleep parameters after sleep deprivation in NPer2 and GPer2 KO mice, both genotypes gained on average 48min and 36min, more sleep in the dark phase under unchallenged conditions. Because locomotion was unchanged, the increase in sleep amount was not due to diminished or altered physical activity. We deduct from this, that either neuronal or astrocytic Per2 contribute on a minor scale to overall sleep need during the active phase in the state of sleep-wake equilibrium. However, when the sleep-wake-ratio set-point is thrown off balance, both neuronal or astrocytic Per2 seem not to be involved in the homeostatic recovery response.

How did the feeding behaviour compare with neuronal and astrocyte mice? Did they change? 

We did not test NPer2 and GPer2 mice in metabolic cages since they did not show a sleep phenotype. Since the rationale for our experiments were to understand the impact of circadian gene Per2 on sleep and sleep regulation, we considered possible underlying metabolic changes coming from this circadian-sleep-interaction perspective. Per se we agree, that it would be very interesting to investigate the impact of lack of Per2 specifically in neurons or astrocytes on metabolism. We are considering testing NPer2 and GPer2 mice for metabolic changes in the future.

Since other glial cells (ex: microglia) are not per2 deleted. So, I recommend changing ‘glial’ to astroglial throughout the manuscript for clarity

We adapted this.

Reviewer 3 Report

The data in this paper seem to be carefully collected, using appropriate methods and statistical tests. The quality of the work is of a high technical standard. However, I believe there are serious problems with the concepts behind the experimental design that greatly reduce the impact of the conclusions that can be reached from this data set.

There seems to be a major flaw in the reasoning behind the conclusion that (lines 531-533) metabolism impacts sleep and peripheral oscillators play a role in sleep regulation. Lines 14-15 in the abstract say they want to investigate clock-driven metabolic contributions to sleep regulation, but this is not what they have done. The per2 whole-body knockouts presumably knock out per2 in all the neurons and glia as well as the periphery. They have compared these whole-body knockouts to knockouts of just the neurons or just the glia. How can the authors eliminate the possibility that the effects seen in whole-body knockouts are not due to the simultaneous knockout of both neurons and glia, and has nothing to do with peripheral oscillators being knocked out? The whole-body knockouts should have been compared to mice with both neuronal and glial per2 knocked out in order to make that conclusion. The experiment suggested in lines 523-524 would also be necessary, knocking out per2 in only peripheral tissues. There are therefore only limited conclusions that can be made from this data set: that single knockouts of neurons or glia have little effect on sleep regulation, and whole-body knockouts have an effect on sleep parameters. From these results it is possible to conclude that per2 contributes to sleep regulation, but further conclusions about whether correct sleep regulation requires per2 in the periphery or requires per2 in both neurons and glia cannot be determined. 

There are other major problems with the manuscript:

1) The introduction is inadequate at setting out the background for the paper. The authors have not adequately explained the rationale for this work. Why was it important to do this research? What specific hypotheses were they testing? How would their experimental design test those hypotheses? There is a general statement in lines 49-51 about the per2 gene being part of the clock and metabolism and so they tested sleep regulation, but what were their predictions for effects? Lines 53-54 talk about communication between neurons and astrocyte, but this was not specifically tested.

2) There is no explanation of why they tested two different whole-body knockouts, both Per2-Brdm and TPer2. Why were both tested? And importantly, how do the authors then explain the large differences between these two knockouts that should have identical phenotypes, particularly the differences in metabolic parameters in Table 1? There are also large differences between the two wild-type strains, and this also requires comments from the authors. Is this a result of the different genetic backgrounds in these strains?

3) Data presentation needs to be improved for the graphs of distributions such as Fig 1C and many others. It is very hard to extract the important conclusions from so many bar height comparisons. Could the authors synthesize these graphs into easy-to-understand metrics, such as skewness of the distribution? The bar graphs could go in the supplement.

Minor problems:

4) Text summarizing the experimental design (lines 72-75) is repeated at the beginning of each section, in lines 194-197, 280-284 and 367-370. Delete the later repeats and just refer to the first description “as previously”.

5) Figures S3 and S4 are reversed – S3 is really S4 and vice versa.

6) Line 168: Should “triggering” be “masking”?

7) Line 261, and line 425: Are those the correct figure citations?

Author Response

Comments and Suggestions for Authors

The data in this paper seem to be carefully collected, using appropriate methods and statistical tests. The quality of the work is of a high technical standard. However, I believe there are serious problems with the concepts behind the experimental design that greatly reduce the impact of the conclusions that can be reached from this data set.

There seems to be a major flaw in the reasoning behind the conclusion that (lines 531-533) metabolism impacts sleep and peripheral oscillators play a role in sleep regulation. Lines 14-15 in the abstract say they want to investigate clock-driven metabolic contributions to sleep regulation, but this is not what they have done. The per2 whole-body knockouts presumably knock out per2 in all the neurons and glia as well as the periphery. They have compared these whole-body knockouts to knockouts of just the neurons or just the glia. How can the authors eliminate the possibility that the effects seen in whole-body knockouts are not due to the simultaneous knockout of both neurons and glia, and has nothing to do with peripheral oscillators being knocked out? The whole-body knockouts should have been compared to mice with both neuronal and glial per2 knocked out in order to make that conclusion. The experiment suggested in lines 523-524 would also be necessary, knocking out per2 in only peripheral tissues. There are therefore only limited conclusions that can be made from this data set: that single knockouts of neurons or glia have little effect on sleep regulation, and whole-body knockouts have an effect on sleep parameters. From these results it is possible to conclude that per2 contributes to sleep regulation, but further conclusions about whether correct sleep regulation requires per2 in the periphery or requires per2 in both neurons and glia cannot be determined. 

We thank the reviewer for this important comment. We agree that a combination of NPer2 and GPer2 may be important which we did not assess nor did we assess deletion of Per2 in the periphery alone (e.g. muscle).

We adapted the discussion and added the sentence: However, another interpretation of our data could be that the sleep phenotypes observed for the whole-body knock-out mice (Per2Brdm and TPer2) stem from the simultaneous knockout of both neurons and glia. Compensatory mechanisms of either neurons or glial cells might be sufficient to rescue the sleep regulation response of the cell-specific knockouts. Therefore, it remains to be seen whether lack of Per2 in neurons and astrocytes combined or Per2 in peripheral organs contributes to the sleep phenotype observed in whole body Per2 knock-out mice.

There are other major problems with the manuscript:

1) The introduction is inadequate at setting out the background for the paper. The authors have not adequately explained the rationale for this work. Why was it important to do this research? What specific hypotheses were they testing? How would their experimental design test those hypotheses? There is a general statement in lines 49-51 about the per2 gene being part of the clock and metabolism and so they tested sleep regulation, but what were their predictions for effects? Lines 53-54 talk about communication between neurons and astrocyte, but this was not specifically tested.

We agree that the reason why we did these experiments were described in a confusing manner. We changed this and the last paragraph of the introduction now reads as follows:

Since the Per2 gene is part of both, the circadian clock mechanism [19] as well as the regulatory mechanism of metabolic processes [20,21], mice with a mutation in the Per2 gene were investigated for its role in sleep regulation [7,22]. The studies indicated, that in these mutants not only the circadian component of sleep was altered but also the homeostatic component appeared to be affected. In these studies, Per2 was deleted in the whole organism not differentiating the role of Per2 in specific cell types. Because sleep is generally viewed as a behavior emerging in the brain and Per2 is expressed in both neurons and astroglia, which metabolically dependent on each other [23], we wanted to test whether neurons (nestin positive cells) or astroglial cells (gfap positive cells), were responsible for the described sleep regulatory function of Per2. Using a non-invasive method we analyzed sleep-wake parameters of total Per2 (TPer2), neuronal Per2 (NPer2) [24] and glial Per2 (GPer2) [25] knock-out (KO) animals and compared them with mice containing a Per2 gene with an in-frame deletion (Per2Brdm) [26].

2) There is no explanation of why they tested two different whole-body knockouts, both Per2-Brdm and TPer2. Why were both tested? And importantly, how do the authors then explain the large differences between these two knockouts that should have identical phenotypes, particularly the differences in metabolic parameters in Table 1? There are also large differences between the two wild-type strains, and this also requires comments from the authors. Is this a result of the different genetic backgrounds in these strains?

We agree with the reviewer that this has not been sufficiently explained. At the beginning these were just controls, because we expected that either the neuronal or the astrocyte specific knock-out would show a similar phenotype. Since the Per2 Brdm KO is in another strain than the NPer2 and GPer2 mice, we used also the TPer2 animals from the same strain as controls. The differences between the TPer2 and the Per2 Brdm KO are related to genetic background differences associated with metabolism, indicating that genetic background and metabolism caused the differences between Per2Brdm and TPer2 mice.

We explain this now at the beginning of the results section:

To examine the role of clock gene Per2 in sleep including but not limited to brain cell specificity, we investigated mice with a mutation (Per2Brdm) or deletion of this gene in all cells of the body (TPer2) or specifically in neurons (NPer2) or glial cells (GPer2), respectively. Due to different Per2 deletion strategies, Per2Brdm and TPer2 mice were not on the same genetic background which may be associated with metabolic differences. To control for genetic background, control animals used in each comparison termed wild-type (wt) were littermates (for NPer2 and GPer2) or corresponding strain wildtypes (mixed SV129/B6 background for Per2Brdm, B6 background for TPer2). Therefore, the wt in each of the following comparisons are not the same but specific controls to the corresponding knock-out (Per2Brdm, TPer2, NPer2 and GPer2). Both total knock-out strains were analysed, because Per2Brdm mutants have been widely used to study the role of Per2 in circadian rhythms and TPer2 mice possess the same genetic background as NPer2 and GPer2 animals. This allowed us to compare the cell-type specific KOs with TPer2 mice and on the other hand we could study potential metabolic differences between the two whole body knock-out strains due to genetic background differences.

3) Data presentation needs to be improved for the graphs of distributions such as Fig 1C and many others. It is very hard to extract the important conclusions from so many bar height comparisons. Could the authors synthesize these graphs into easy-to-understand metrics, such as skewness of the distribution? The bar graphs could go in the supplement.

We prefer to show the least processed data. We agree that this is not always the easiest way of visualization but the most direct one. As molecular biologists we want to see the original unprocessed data and make from those interpretations. I guess this is an issue of scientific culture in different fields. Therefore, we leave the presentation of data as they are.

Minor problems:

4) Text summarizing the experimental design (lines 72-75) is repeated at the beginning of each section, in lines 194-197, 280-284 and 367-370. Delete the later repeats and just refer to the first description “as previously”.

We have followed this suggestion.

5) Figures S3 and S4 are reversed – S3 is really S4 and vice versa.

We corrected this.

6) Line 168: Should “triggering” be “masking”?

We wrote now triggering or de-masking. Light can mask activity, therefore lifting that condition no longer suppresses activity (hence “de-masks”).

7) Line 261, and line 425: Are those the correct figure citations?

Corrected:

  • For line 261: S2B, S4B
  • For line 425: In the 2h of the dark phase (ZT13 to ZT15, blue squares in Fig. 4B), sleep bout duration and total sleep were similar in GPer2 and control mice, pre-SD (hours 13-14 in figure 4B) as well as post-SD (hours 38-39 in figure 4B) (Figs. 4H, S5).

Reviewer 4 Report

This manuscript nicely describes the sleep-relevant effects of Per2-loss from neurons, astrocytes, and all clock cells, and presents a comparison of the phenotypes.

There are some minor discrepancy between Per2Brdm and TPer2-KO mice sleep phenotypes; I would request the authors to point out the possible origin of these.

Evidently, neither neuronal nor astrocytic Per2 expression is crucial for sleep. Could the authors suggest which peripheral clock(s), they think, is involved?

I do find that the implication of metabolism in the sleep-relevant phenotypes that the authors are arguing is slightly weak. Removing this at least from the title of the article would be my advice.

Author Response

This manuscript nicely describes the sleep-relevant effects of Per2-loss from neurons, astrocytes, and all clock cells, and presents a comparison of the phenotypes.

There are some minor discrepancies between Per2Brdm and TPer2-KO mice sleep phenotypes; I would request the authors to point out the possible origin of these.

We agree with the reviewer that this has not been sufficiently explained. The differences between the TPer2 and the Per2 Brdm KO are related to genetic background differences, indicating that genetic background potentially related to metabolism caused the differences between Per2Brdm and TPer2 mice.

We explain this now at the beginning of the results section:

To examine the role of clock gene Per2 in sleep including but not limited to brain cell specificity, we investigated mice with a mutation (Per2Brdm) or deletion of this gene in all cells of the body (TPer2) or specifically in neurons (NPer2) or glial cells (GPer2), respectively. Due to different Per2 deletion strategies, Per2Brdm and TPer2 mice were not on the same genetic background which may be associated with metabolic differences. To control for genetic background, control animals used in each comparison termed wild-type (wt) were littermates (for NPer2 and GPer2) or corresponding strain wildtypes (mixed SV129/B6 background for Per2Brdm, B6 background for TPer2). Therefore, the wt in each of the following comparisons are not the same but specific controls to the corresponding knock-out (Per2Brdm, TPer2, NPer2 and GPer2). Both total knock-out strains were analysed, because Per2Brdm mutants have been widely used to study the role of Per2 in circadian rhythms and TPer2 mice possess the same genetic background as NPer2 and GPer2 animals. This allowed us to compare the cell-type specific KOs with TPer2 mice and on the other hand we could study potential metabolic differences between the two whole body knock-out strains due to genetic background differences.

Evidently, neither neuronal nor astrocytic Per2 expression is crucial for sleep. Could the authors suggest which peripheral clock(s), they think, is involved?

Of course, this would be very speculative. Nonetheless, we give the example of clock gene Bmal1 in muscle:

“This would suggest that sleep is a systemic phenomenon that is not wholy dependent on the brain only and that peripheral clocks may be involved in sleep regulation. In support of this view is a study which showed that the clock gene Bmal1 in skeletal muscle regulates sleep [30]. Restoration of Bmal1 expression in the brain of Bmal1 whole body knock-out (Bmal1 KO) animals did not restore the sleep phenotype of the Bmal1KO strain, while restoring the expression of this gene in skeletal-muscle did [30]. Therefore, it would be interesting to test whether deletion of Per2 in peripheral organs such as the muscle only would be sufficient to affect sleep.”

I do find that the implication of metabolism in the sleep-relevant phenotypes that the authors are arguing is slightly weak. Removing this at least from the title of the article would be my advice.

We agree.

New title:

Lack of the Circadian Clock Gene Per2 in Neurons or Astroglia does not Affect Sleep in Response to Sleep Deprivation

Round 2

Reviewer 2 Report

The authors have satisfactorily addressed all the concerns raised. So, I would recommend 'accept in present form'

Author Response

We thank the reviewer for the time to go through our manuscript again.

Reviewer 3 Report

The authors have adequately dealt with the criticisms of the reviewers. I do have a suggestion for a better title:

Deletion of Per2 in the whole body, but not in neurons or glia, affects sleep in response to sleep deprivation

Author Response

We thank the reviewer for his suggestion to change the title. We think this is an excellent idea and makes the paper more appealing. Thank you!

The title is now:

Deletion of the Circadian Clock Gene Per2 in the Whole Body, but not in Neurons or Astroglia, Affects Sleep in Response to Sleep Deprivation